# Vaccine-induced COVID-19 mimicry syndrome

Eric Kowarz[1†], Lea Krutzke[2†], Marius Külp[1], Patrick Streb[1], Patrizia Larghero[1], Jennifer Reis[1], Silvia Bracharz[1], Tatjana Engler[2], Stefan Kochanek[2], Rolf Marschalek[1]*

[1]Institute of Pharmaceutical Biology/DCAL, Goethe-University of Frankfurt, Frankfurt am Main, Germany; [2]Department of Gene Therapy, Ulm University, Ulm, Germany

**\*For correspondence:**
Rolf.Marschalek@em.uni-frankfurt.de

[†]These authors contributed equally to this work

**Competing interest:** The authors declare that no competing interests exist.

**Abstract** To fight the COVID-19 pandemic caused by the RNA virus SARS-CoV-2, a global vaccination campaign is in progress to achieve the immunization of billions of people mainly with adenoviral vector- or mRNA-based vaccines, all of which encode the SARS-CoV-2 Spike protein. In some rare cases, cerebral venous sinus thromboses (CVST) have been reported as a severe side effect occurring 4–14 days after the first vaccination and were often accompanied by thrombocytopenia. Besides CVST, splanchnic vein thromboses (SVT) and other thromboembolic events have been observed. These events only occurred following vaccination with adenoviral vector-based vaccines but not following vaccination with mRNA-based vaccines. Meanwhile, scientists have proposed an immune-based pathomechanism and the condition has been coined vaccine-induced immune thrombotic thrombocytopenia (VITT). Here, we describe an unexpected mechanism that could explain thromboembolic events occurring with DNA-based but not with RNA-based vaccines. We show that DNA-encoded mRNA coding for Spike protein can be spliced in a way that the transmembrane anchor of Spike is lost, so that nearly full-length Spike is secreted from cells. Secreted Spike variants could potentially initiate severe side effects when binding to cells via the ACE2 receptor. Avoiding such splicing events should become part of a rational vaccine design to increase safety of prospective vaccines.

## Editor's evaluation

In this manuscript, the authors provide evidence for the occurrence of splice reactions in adenovirus-based current vaccines resulting in the secretion of truncated Spike variants providing a potential mechanisms underlying thromboembolic events that have been reported for DNA-based but not for RNA-based COVID-19 vaccines.

## Introduction

The COVID-19 pandemic, starting in the last months of 2019 in Wuhan (China) and caused by the RNA virus SARS-CoV-2, so far (January 4, 2022) has resulted in more than 293.0 million (Mio) infections and more than 5.46 Mio deaths (data source: https://www.worldometers.info/coronavirus/). The world has faced the fastest development and production of vaccines ever, resulting in a total of four vaccines that have received conditional market authorizations by the regulatory bodies: the mRNA vaccines from BioNTech/Pfizer (BTN162b2/Comirnaty; EMA: 21.12.20) and Moderna (mRNA-1273/Spikevax; EMA: 6.1.21), as well as the adenoviral vector-based vaccines from AstraZeneca (AZD1222/ChAdOx1-S/Vaxzevria; EMA: 29.1.21) and Janssen (Ad26.COV2.S; EMA: 11.3.21). All four vaccines encode slightly different forms of the SARS-CoV-2 Spike glycoprotein, which mediates virus binding to the host cell membrane and entry via ACE2 and TMPRSS2, respectively (*Hoffmann et al., 2020*).

**Table 1.** Safety data from the German Paul-Ehrlich Institute (PEI), dated from December 23, 2021. All data for all four vaccines were directly retrieved from the security report. Data for number of all vaccinations, total cases with complications, anaphyllaxie cases, myo/pericarditis cases, cases with thromboses associated with thrombocytopenia (TTS), cases with thromboses combined with immunothrombocytopenia (ITP), and cases with Guillain-Barré syndrome (GBS) were listed. In addition, for each category the case numbers per 1 million (Mio) injections are given (e.g. the absolute risk for the development of myo/pericarditis after a Biontech vaccination is 12.9 cases per Mio injections). All red numbers differ significantly from the numbers of the other vaccines.

| Company | Biontech/Pfizer | Moderna | Oxford/AZ | Janssen/J&J |
|---|---|---|---|---|
| #Vaccinations | 96.606.131 | 10.576–131 | 12.703.030 | 3.462.557 |
| #Complications | 113.792 | 28.289 | 46.325 | 7.758 |
| **Cases per Mio inj.** | 1.178 | 2.675 | **3.647** | 2.241 |
| Anaphyllaxie | 550 | 55 | 101 | 10 |
| **Cases per Mio inj.** | 5,7 | 5.2 | 8.0 | 2,9 |
| Myo/pericarditis | 1245 | 309 | 0 | 0 |
| **Cases per Mio inj.** | **12.9** | **29.2** | 0 | 0 |
| TTS | 36 | 5 | 200 | 24 |
| **Cases per Mio inj.** | 0.4 | 0.5 | **15.7** | **6.9** |
| ITP | 314 | 28 | 269 | 23 |
| **Cases per Mio inj.** | 3.3 | 2.6 | **21.2** | 6.6 |
| GBS | 140 | 14 | 112 | 48 |
| **Cases per Mio inj.** | 1.4 | 1.3 | **8.8** | **13.9** |
| Death cases | 295 | 20 | 201 | 21 |
| **Cases per Mio inj.** | 3.1 | 1.9 | **15.8** | 6.1 |

While vaccination with BioNTech and Moderna mostly causes only mild and typical immediate vaccination side effects, severe side effects thromboses (ITP) or thrombosis with thrombocytopenia syndrome (TTS) were first observed with Vaxzevria in Europe (in March 2021). The most severe side effects entailed rare events of thrombocytopenia combined with cerebral venous sinus thrombosis (CVST). The higher than expected occurrence of CVST led to a rapid suspension of the Vaxzevria vaccination campaign in several European countries (March 11–14, 2021: Denmark, several northern European countries, Thailand, Ireland; March 15, 2021: Germany, Italy, France, and Spain).

The latest data from the security surveillance of the Paul-Ehrlich-Institute in Germany (report dated from December 23, 2021; https://www.pei.de/DE/newsroom/dossier/coronavirus/sicherheitsbericht-covid-19-impfstoffe-aktuell.html) revealed still a significant risk increase for the AstraZeneca vaccine compared to the other vaccines (see *Table 1*). Thrombocytopenia and immunothrombocytopenia (ITP) have been observed in 314 cases after 96.6 Mio vaccination doses with Comirnaty (3.3 cases per Mio injections), in 28 cases with Spikevax after 10.5 Mio vaccination doses with Comirnaty (2.6 cases per Mio injections), in 269 with Vaxzevria after 12.7 Mio vaccination doses cases (21.2 cases per Mio injections), and in 23 cases with Ad26.COV2.S after 3.4 Mio vaccination doses cases (6.6 cases per Mio injections). Thrombosis-related death cases for these four vaccines were 74, 6, 81, and 12, respectively. This indicates that thromboembolic events associated with case mortality were in the range of 21.4–52% (see *Table 1*).

TTS has been observed for these four vaccines in 36, 5, 200, and 24 patients, indicating a relative risk of 0.4, 0.5, 15.7, and 6.9 cases per Mio injections. These thromboembolic events in combination with thrombocytopenia share similarities with another condition, 'heparin-induced thrombocytopenia' (HIT). The term 'vaccine-induced immune thrombotic thrombocytopenia' (VITT) was coined for this vaccine-induced condition, after finding out that autoantibodies against the platelet factor 4 (PF4) could be causally linked to it (*Greinacher et al., 2021b*; *Greinacher et al., 2021a*). However,

despite intensive research activities by many laboratories, the underlying initiating mechanisms for the observed side events were not yet fully understood.

At the molecular delivery mechanism of the approved vaccines, there are fundamental differences between mRNA- and adenoviral vector-based vaccines. The mRNA vaccines encode, as the naming implies, the Spike sequence as a modified RNA sequence encapsulated in lipid nanoparticles. Upon intramuscular injection, the nanoparticles are taken up by the cells, where the mRNA cargo is released into the cytosol. Since both mRNA vaccines encode a signal peptide, the process of protein translation occurs at the rough endoplasmatic reticulum (ER) and results in membrane-anchored Spike proteins. Further post-translational modifications occur in the ER and Golgi apparatus (glycosylation) from where the Spike protein trimer is transported in vesicles to the outer cell membrane.

In contrast, adenoviral vector-based vaccines deliver the Spike sequence as a codon-optimized DNA sequence instead of a RNA sequence. Vaxzevria is based on a genetically modified chimpanzee adenovirus type Y25, while the Janssen vaccine is based on a human adenovirus type 26 (HAdV-D26), both of which are replication incompetent. Upon transduction of the host cell and disassembly of the vector particle, the linear double-stranded DNA is released and enters the nucleus. Subsequently, viral DNA-encoded genes become transcribed using the host cell transcription machinery (*Doerfler and Böhm, 2003*). In the nucleus and before the mRNA is exported to the cytosol for translation (see above), the resulting RNA molecule is subject to the same post-transcriptional RNA processing machinery as endogenous primary transcripts, including the possibility of splicing.

SARS-CoV-2 is a single-stranded RNA virus of the betacoronavirus family that naturally has a purely cytoplasmic replication cycle. Therefore, coronaviruses have not been exposed to any evolutionary pressure that would have an impact on biological processes taking place in the nucleus. Almost all nuclear-encoded genes contain introns that become eliminated from precursor RNA transcripts by the splice machinery (*Lee and Rio, 2015*). Defined RNA consensus sequences are recognized by RNA/protein complexes (spliceosome), which consist of nearly 100 proteins and specific U-RNAs. Noteworthy, splice events are influenced by exonic splice enhancer sequences and exonic splice silencer sequences, which bind splice-activating or -suppressing proteins. During this evolutionary conserved splice process, a so-called 'branch A' nucleotide is performing a nucleophilic attack at a G•G-dinucleotide of the 5'-splice consensus sequence G•GUNNGU. The resulting 3'-OH of the final exonic G nucleotide then makes a second nucleophilic attack at the G•G-dinucleotide of the 3'-splice consensus sequence YYYYYCAG•G. To this end, two exons become fused, and the intronic sequence is being eliminated. For each gene, this process guarantees that all introns flanked by SD and SA sites are eliminated in order to produce the final mRNA that encodes a specific protein-coding sequence.

When using cloned cDNA in molecular biology or biotechnology, no intronic sequences are usually present, because cDNA synthesis is made from already spliced mRNA molecules. However, in case of the SARS-CoV-2 virus, the cloned cDNA for the Spike open reading frame was not coming from a nuclear-encoded gene, rather from an RNA virus that replicates solely in the cytosol of infected cells. Therefore, such a cDNA exhibits potential SD and SA sites just by chance (~130 cryptic splice signal sequences in the total SARS-CoV-2 virus sequence), because no negative selection has ever taken place during the evolution of coronavirus to eliminate such sequences. Therefore, we hypothesized that nuclear splicing of DNA-encoded Spike RNA sequences might result in spliced, and thus, truncated Spike protein variants with unknown fate and function. As mentioned above, this type of splicing would be quite artificial, because an RNA virus does per se not contain any intronic sequences in its various open reading frames. Thus, all such splicing events are derived from the so-called cryptic splice sites. This is unrelated to the mechanism of alternative splicing, which could be functionally attributed to certain genes of our genome. Cryptic splicing occurs only if strong splice sites are present, and thus, these cryptic splice sites will be used equally in different cell types or tissues.

To address this question of cryptic splicing, we analyzed nuclearly transcribed wildtype and codon-optimized Spike genes in a specially designed splice reporter vector system. We also investigated splicing of Spike RNAs following transduction of representative cells with a Spike-expressing human adenovirus type 5 vector and the two approved adenoviral vector-based vaccines ChAdOx1-S and Ad26.COV2.S. Here, we show that splicing of DNA-encoded SARS-CoV-2 Spike RNA molecules takes place at positions as they were predicted by in silico analysis. These splice events resulted in Spike protein variants that lack the transmembrane (TM) anchor, thus may become secreted due to the nature of the secretory pathway. Codon optimization performed to enhance expression levels further

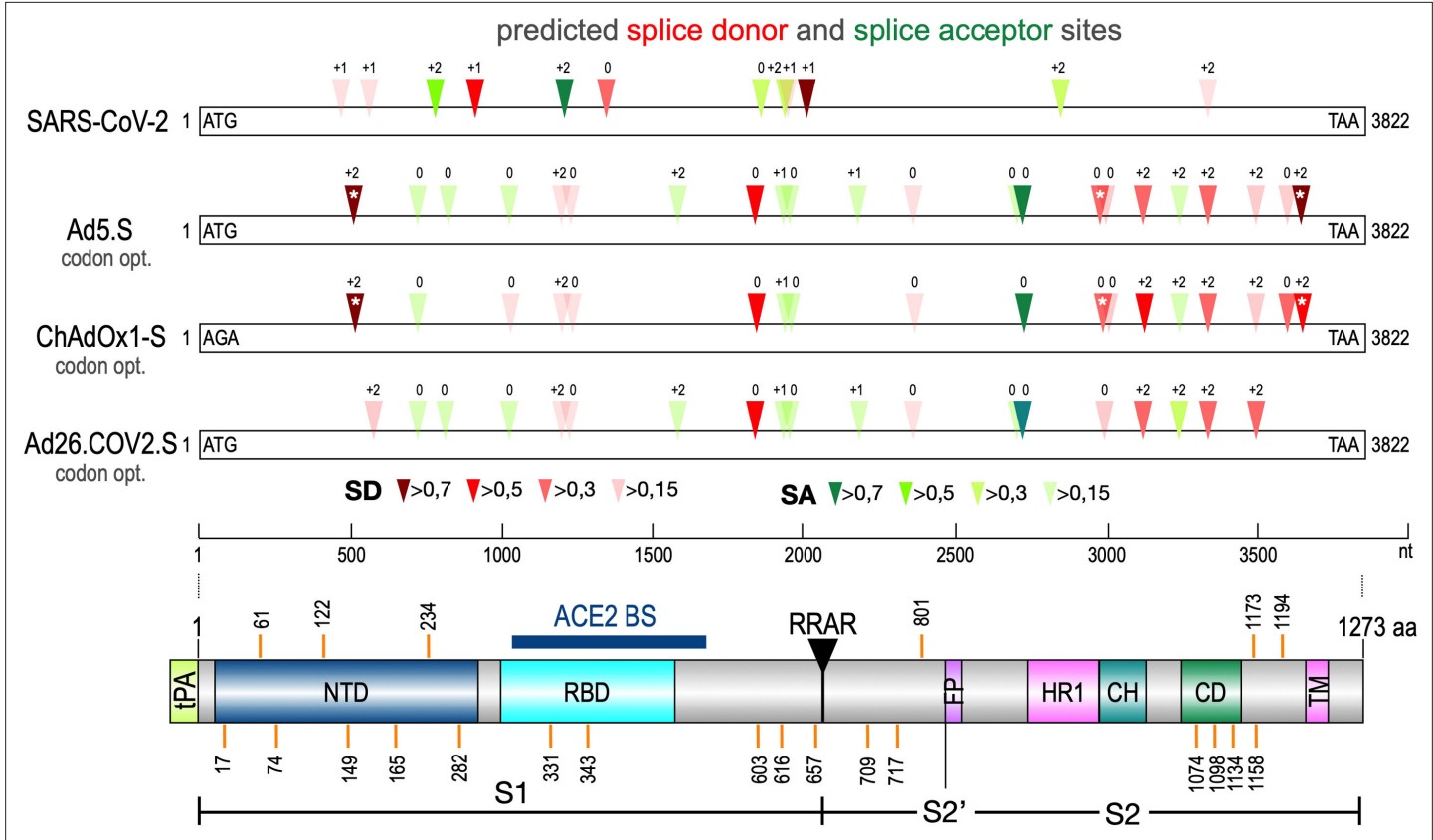

**Figure 1.** Splice site prediction within the Spike open reading frames. Splice site prediction was carried out by using SpliceRover. Splice donor sites are given in red, splice acceptor sites in green. SpliceRover calculates splice sites with a score between 0 and 1, but only splice sites with >0.15 were displayed. Sites were displayed in four colors as indicated. Splice sites were numbered with '0', '+1', or '+2', to indicate how the open reading frame is disrupted. Therefore, all splice events between splice sites with identical numbers will be in-frame, while all splice reactions between unequal numbers will result in out-of-frame fusions. Below, the protein domain structure of Spike is displayed to explain, which domains are being deleted by splice events (NTD: N-terminal domain, RBD: (ACE2) receptor-binding domain, RRAR: furin cleavage site, HR1: heptad repeat1, CH: central helix, CD: connector domain, TM: transmembrane domain). White asterisks mark three splice donor sites that are present in the codon-optimized Spike reading frame of the Vaxzevria but not of the Janssen vaccine.

augmented the abundance of splicing and was most prominently observed for the ChAdOx1-S vector. In addition, we analyzed the production of Spike protein and its secretion to the outside medium. Implications of our findings for the safety of vector-based vaccines and future consequences for the design of vector-based vaccines for RNA viruses will be discussed.

## Results

### In silico analyses reveal potential splice sites in adenoviral vector-encoded Spike sequences

The Spike open reading frame of the Wuhan SARS-CoV-2 isolate was tested for potential splice sites by using the available SpliceRover online tool (*Zuallaert et al., 2018*). In addition, we used the online Alternative Splice Site Predictor (ASSP) tool to confirm the predicted splice sites (data not shown). Both servers identified several dozens of potential splice sites in the 3822 nucleotide long open reading frame of the Spike gene. Therefore, we decided to filter these hits by their individual scores to identify those, which were used with a higher likelihood (wildtype Spike gene: six splice donor [SD] sites, five splice acceptor [SA] sites, see *Figure 1* and *Table 2A*).

Similarly, we performed the same in silico analysis also for codon-optimized Spike open reading frames in three different adenovirus vector systems. First, these were an experimental and codon-optimized variant of the original Wuhan sequence, cloned into a human adenovirus type five vector

**Table 2.** Splice donor site prediction.

The open reading frame of the SARS-CoV-2 Spike protein was analyzed by the Splice Site Predictor online tool SpliceRover (http://bioit2.irc.ugent.be/rover/splicerover). This online tool allows prediction of splice donor and splice acceptor sites. Predicted splice donor sites are provided in *Table 1* for the SARS-CoV-2 wildtype Spike gene (A), for the codon-optimized Spike genes derived from the experimental Ad5.S vector, and the FDA/EMA-approved vector-based vaccines from AstraZeneca (Vaxzevria, ChAdOx1-S) and Janssen/J&J (Ad26.COV2.S), respectively. The amino acid coordinates of Spike protein domains S1, S2, and the minimal ACE2-binding domain are indicated.

| Position | Potential splice donor sites | Score | Type |
|---|---|---|---|
| **A: Splice donor site prediction in wildtype Spike ORF** | | | |
| 454–473 | TGGATGGAAA•GTGAGTTCAG | 0.187 | +1 |
| 541–560 | GGAAAACAGG•GTAATTTCAA | 0.151 | +1 |
| 894–911 | GAAACAAAGT•GTACGTTGAA | 0.565 | +1 |
| 1323–1342 | TGATTCTAAG•GTTGGTGGTA | 0.357 | 0 |
| 1906–1925 | TATTCTACAG•GTTCTATTFT | 0.274 | +1 |
| 1996–2015 | ATTGGTGCAG•GTATATGCGC | 0.707 | +1 |
| 3296–3317 | GCACACACTG•GTTTGTAACA | 0.160 | +2 |
| Consensus | MAG•GTNNGTG | | |
| **B: Splice donor site prediction in codon-optimized Spike ORF of Ad5** | | | |
| 497–516 | GCACCTTCGA•GTACGTGTCC | 0.987 | +2 |
| 1175–1194 | TCACAAACGT•GTACGCCGAC | 0.187 | +2 |
| 1209–1228 | GGGAGATGAA•GTGCGGCAGA | 0.162 | 0 |
| 1812–1831 | CTCCAACCAG•GTGGCCGTGC | 0.540 | 0 |
| 2331–2350 | CACCCAAGAG•GTGTTCGCCC | 0.204 | 0 |
| 2949–2968 | ACTGGACAAG•GTGGAAGCCG | 0.318 | 0 |
| 2961–2980 | GGAAGCCGAG•GTGCAGATCG | 0.151 | 0 |
| 3083–3102 | AGATGTCTGA•GTGTGTGCTG | 0.318 | +2 |
| 3296–3315 | GCACCCATTG•GTTCGTGACC | 0.388 | +2 |
| 3452–3471 | AACTGGATAA•GTACTTTAAG | 0.443 | +2 |
| 3555–3574 | GCTGAACGAG•GTGGCCAAGA | 0.217 | 0 |
| 3605–3624 | AACTGGGGAA•GTACGAGCAG | 0.800 | +2 |
| **C: Splice donor site prediction in codon-optimized Spike ORF of ChAdOx1-S** | | | |
| 497–516 | GCACCTTCGA•GTACGTGTCC | 0.986 | +2 |
| 1012–1030 | TTCGGCGAG• GTGTTCAATG | 0.266 | 0 |
| 1175–1194 | TCACAAACGT•GTACGCCGAC | 0.177 | +2 |
| 1209–1228 | GGGAGATGAA•GTGCGGCAGA | 0.188 | 0 |
| 1812–1831 | CTCCAACCAG•GTGGCCGTGC | 0.541 | 0 |
| 2331–2350 | CACCCAAGAG•GTGTTCGCCC | 0.215 | 0 |
| 2949–2968 | ACTGGACAAG•GTGGAAGCCG | 0.398 | 0 |
| 2961–2980 | GGAAGCCGAG•GTGCAGATCG | 0.273 | 0 |
| 3083–3102 | AGATGTCTGA•GTGTGTGCTG | 0.503 | +2 |
| 3296–3315 | GCACCCATTG•GTTCGTGACC | 0.381 | +2 |

*Table 2 continued on next page*

*Table 2 continued*

| Position | Potential splice donor sites | Score | Type |
|----------|------------------------------|-------|------|
| 3452–3471 | AACTGGATAA•GTACTTTAAG | 0.245 | +2 |
| 3555–3574 | GCTGAACGAG•GTGGCCAAGA | 0.310 | 0 |
| 3605–3624 | AACTGGGGAA•GTACGAGCAG | 0.686 | +2 |
| D: Splice donor site prediction in codon-optimized Spike ORF of Ad26.COV2.S | | | |
| 563–582 | ACCTGCGCGA•GTTCGTGTTC | 0.150 | +2 |
| 1175–1194 | TCACAAACGT•GTACGCCGAC | 0.180 | +2 |
| 1209–1228 | GGGAGATGAA•GTGCGGCAGA | 0.207 | 0 |
| 1812–1831 | CAGCAATCAG•GTGGCAGTGC | 0.547 | 0 |
| 2331–2350 | CACCCAAGAG•GTGTTCGCCC | 0.202 | 0 |
| 2961–2980 | TGAGGCCGAG•GTGCAGATCG | 0.194 | 0 |
| 3083–3102 | AGATGTCTGA•GTGTGTGCTG | 0.495 | +2 |
| 3296–3315 | GCACCCATTG•GTTCGTGACA | 0.392 | +2 |
| 3452–3471 | AACTGGACAA•GTACTTTAAG | 0.436 | +2 |

**S1 ectodomain:** encoded by nucleotides 1–2049 (aa **1–683**)

**S2 domain:** encoded by nucleotides 2050–3822 (aa **684–1273**)

**ACE2-binding domain:** encoded by nucleotides 998–1720 (aa **319–551**)

backbone (Ad5.S). Second, the Spike sequence of ChAdOx1-S was derived by sequencing of adenoviral DNA isolated from Vaxzevria, while, third, the Spike sequence of Ad26.COV2.S was obtained directly from Janssen (kindly provided by Roland Zahn). Analyses revealed that codon optimization resulted in an increased number of potential splice sites in all three adenoviral-encoded Spike sequences (Ad5.S: 12 SD sites, 10 SA sites; ChAdOx1-S: 13 SD and 5 SA sites, Ad26.COV2.S: 9 SD and 10 SA sites, see *Table 2B–D*) compared to the open reading frame of the wildtype Wuhan SARS-CoV-2 Spike sequence.

## Significant splicing of codon-optimized Spike sequences

First, we analyzed potential splice events within the open reading frame of wildtype Spike, which contains, according to our prediction, six potential SD sites. Since these splice signal sequences do not occur like in normal genes where every SD site is located upstream of a SA site, both flanking an intronic sequence such SD sites are 'unsaturated' and usually give rise to cryptic splice reactions in *cis* or in *trans* to potential SA (*Kowarz et al., 2011*). To this end, we generated three splice traps in which the Spike sequence was inserted upstream of a perfect SA site, which again was fused in three different reading frames (0, +1, +2) to a full-length Luciferase gene without start codon. In this system Luciferase activity is only detected if splicing occurs. Respective constructs were cloned into a sleeping beauty transposon vector (pSBbi::Spike-Luc-0/+1/+2 GP) and used for stable transfection of HEK293T cells.

Analyses of all three splice trap constructs in Luciferase assay revealed that the '+2 construct' exhibited the strongest intracellular Luciferase activity (1291 relative light units), while the '0 construct' resulted in a much lower activity (341 relative light units). The activity determined for the +1 construct was near the values for non-transfected HEK293T cells, which served as negative controls (see *Figure 2*, panel I.). We also analyzed the medium of the cultured cells and observed that small amounts of Luciferase activity were present also extracellularly. This amount increased over time and mimicked relative ratios of the three constructs observed intracellularly (see *Figure 2*, panels II, III). Due to the design of the three splice traps, Luciferase activity can only be detected when splicing occurs. Thus, these initial Luciferase experiments confirmed that splicing of the Spike sequences was taking place. If splicing occurs between an SD in the Spike open reading frame and the SA in the splice

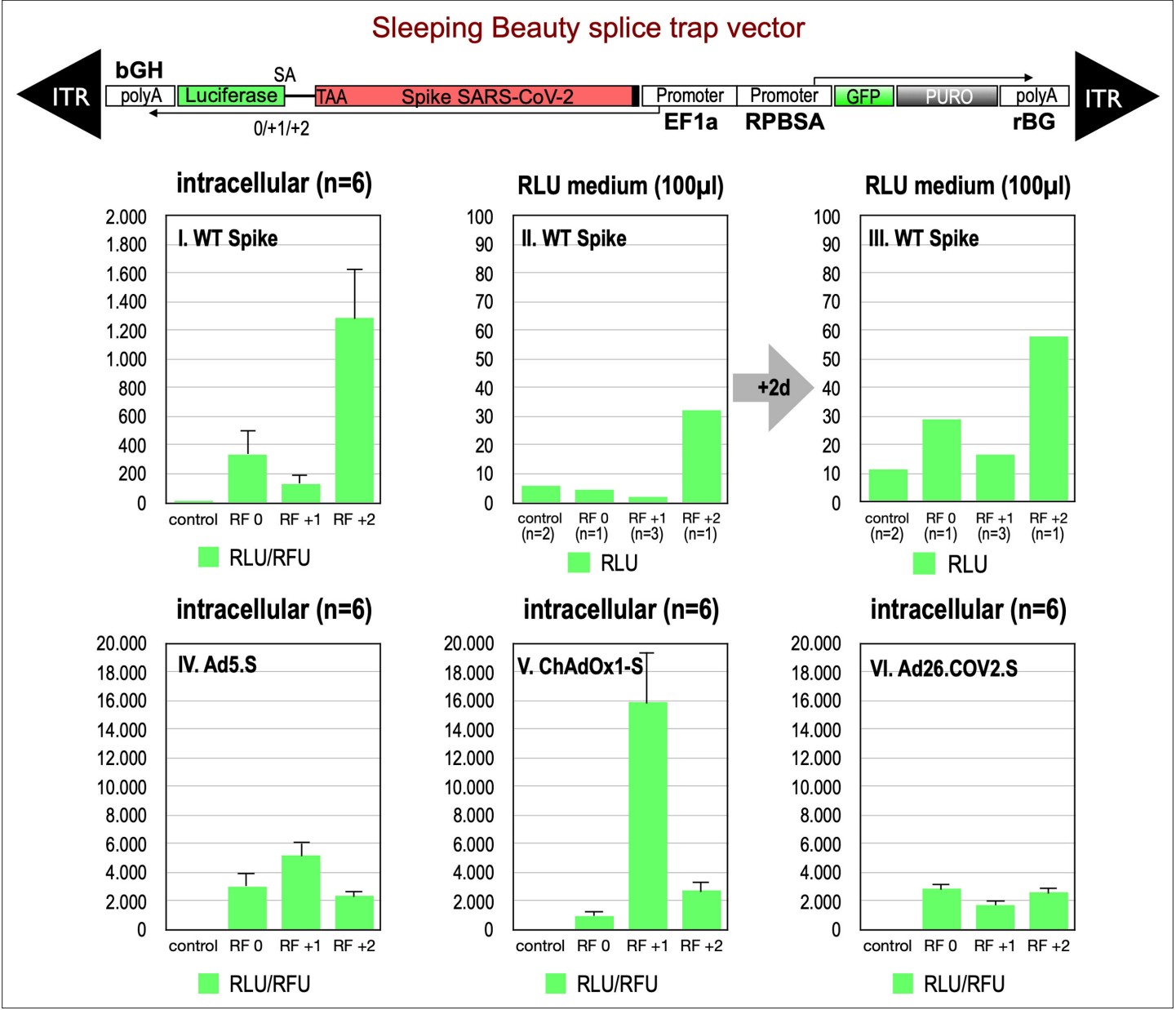

**Figure 2.** Splice trap Luciferase experiments in HEK293T cells. The three different splice traps were cloned into the pSBbi-GP vector. This vector encodes two polycistronic transcripts, one encoding GFP and Puromycin resistance (from left to right), while the other encodes the full-length Spike gene connected to an artificial intron and an ATG-deleted Luciferase gene. Splice events to the Luciferase transcript will result in 'in-frame' and 'out-of-frame' Spike-Luciferase fusion transcripts. HEK293T cells were stably transfected with the constructs and resulting Luciferase activity was measured. RLU: relative light units; RFU: relative fluorescence units. Middle panels I–III: The three different splice trap constructs encode the wildtype Spike gene. Luciferase expression was measured intracellularly and extracellularly 3-day post-transfection. Lower panels IV–VI: The three different splice trap constructs encode the codon-optimized Spike sequences derived from either the adenoviral vector Ad5.S, ChAdOx1-S, or Ad26.COV2.S. Luciferase expression was measured intracellularly.

trap, the C-terminally located TM domain of the Spike protein serving as the membrane anchor will get lost. Thus, resulting truncated Spike proteins may become secreted.

We also cloned the three codon-optimized Spike genes of the adenoviral vectors (Ad5.S, ChAdOx1-S, Ad26.COV2.S) into the three versions of the Luciferase splice trap (0/+1/+2) and analyzed the production of Spike-Luciferase fusion proteins. Here, we only analyzed intracellular Luciferase activity (see *Figure 2*, panels IV–VI). All three codon-optimized Spike genes derived from the different adenoviral vectors displayed a strong production of Spike-Luciferase fusion proteins that was also

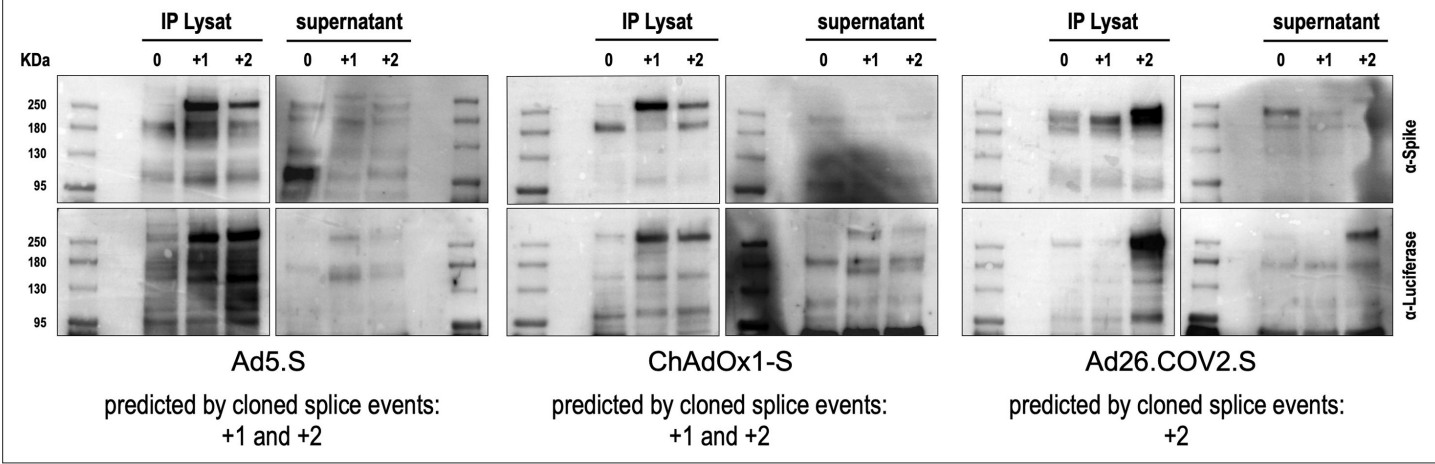

**Figure 3.** Splice trap Western blot experiments in HEK293T cells. Spike and Spike-fusion proteins were concentrated with the help of immunoprecipitation beads from all cell lysates and supernatants of HEK293T cells stably transfected with the splice trap constructs encoding the codon-optimized Spike sequences derived from the adenoviral vectors Ad5.S, ChAdOx1-S, or Ad26.COV2.S. Proteins bound to these beads were used to perform Western blot experiments, which are displayed. Bead eluates were separated by SDS-PAGE, blotted, and proteins were detected with anti-Spike or anti-Luciferase antibodies.

enhanced compared to wildtype Spike sequence constructs. Thus, all codon-optimized Spike genes displayed a higher splicing activity than the wildtype Spike gene sequence. The highest activity of 15,840 relative light units was detected with the reading frame +1 of the ChAdOx1-S construct. The lowest level of Luciferase activity was found in the Ad26.COV2.S construct.

We also performed Western blot experiments to determine steady-state levels of intracellularly produced adenovirus vector-encoded Spike protein variants and Spike-Luciferase fusion proteins. Cell lysates and supernatants of cell lines were used for Spike-immunoprecipitation experiments using protein G beads and anti-FLAG antibodies for capturing. Subsequently, samples were analyzed by Western blot analysis using anti-Spike and anti-Luciferase antibodies. As shown in *Figure 3*, for all analyzed Spike splice traps strong signals of proteins with sizes of 95–250 kDa were detected in cell lysates with both antibodies, confirming Spike-Luciferase fusion proteins of different lengths. Cell lysate and medium supernatant of each investigated cell line displayed several protein bands migrating at molecular weights as predicted by the major splice events in each of the investigated cell lines (Ad5.S 195.1 and 195.3 kDa; ChAdOX1-S 195.1 and 195.3 kDa; Ad26.COV2.S 196.4 kDa; all without glycosylation).

## Verification of various splice products derived from wildtype and codon-optimized Spike open reading frames

To validate these findings at the molecular level, we analyzed RNA from cell lines transfected with the three wildtype Spike-encoding splice trap constructs. Potential splice events were visualized by RT-PCR experiments using a set of four Spike primers pointing 3'-prime against one Luciferase primer pointing 5'-prime. All deviating PCR bands were analyzed by Sanger sequencing (see *Figure 4*, agarose gels). As shown in *Figure 4*, we observed several splice reactions from predicted splice sites (e.g. SD1-3), but also internal splice reactions that lead to frame-shift events. We also found two splice events downstream of the ACE2-binding domain resulting again in a fusion between Spike and Luciferase.

To exclude that observed occurring splicing of the Spike mRNA was an artifact due to the use of artificial splice trap constructs with a very potent SA site at the 5'-end of the Luciferase reading frame, we repeated the analysis using the actual adenoviral vectors Ad5.S, ChAdOx1-S, and Ad26.COV2.S. Here, the Spike sequence is embedded in the adenoviral DNA genome, directly upstream of the adenoviral pIX. Being part of the viral capsid, pIX has mainly structural functions as a 'cement' protein increasing stability of the viral particle, but also has additional functions in the viral life cycle (*Parks, 2005*). The 5'-non-coding sequence of the pIX transcription unit of the three different adenovirus types contains a highly conserved SA site behind the pIX promoter and upstream of the pIX ATG start

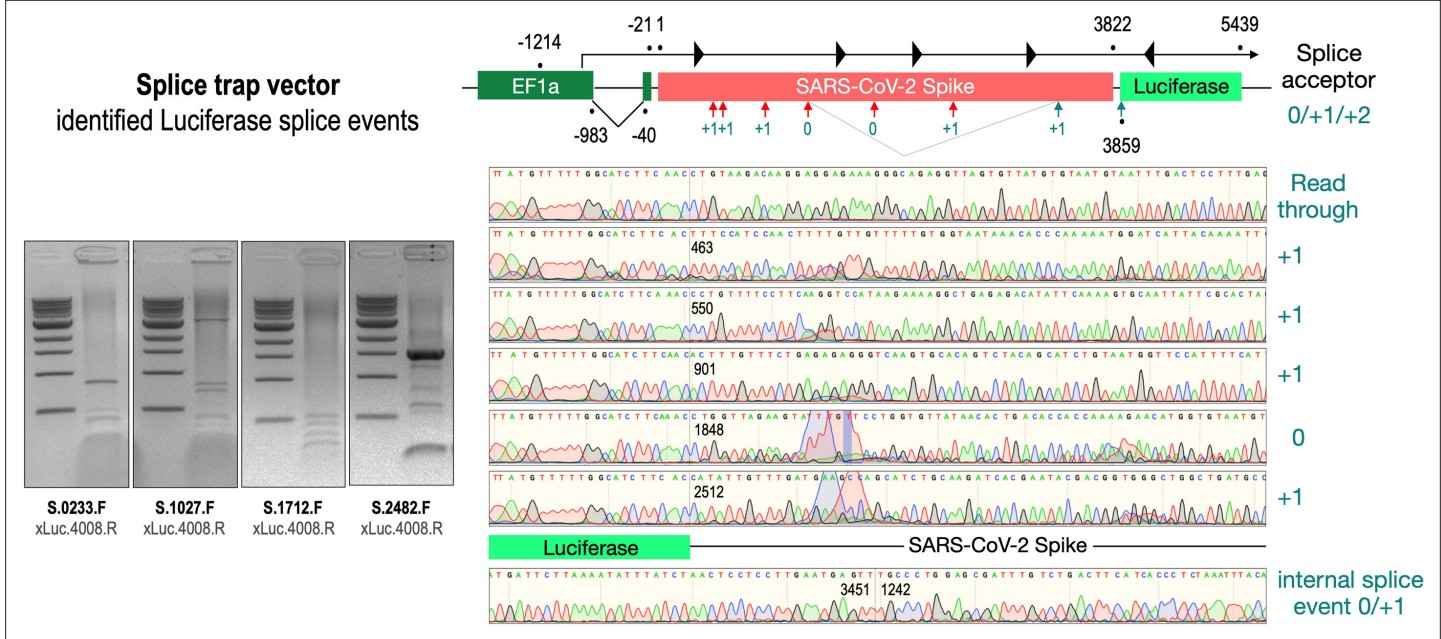

**Figure 4.** Splice events after transfection of HEK293T cells. The wildtype SARS-CoV-2 Spike gene – cloned into the three splice trap constructs – was stably transfected in HEK293T cells, RNA was isolated and investigated by RT-PCR experiments for splicing events. Appropriate PCR bands were cut out from the gel and either directly sequenced or cloned for sequencing. The major splice events discovered are displayed.

codon (Ad5.S pIX at nucleotide –14; ChAdOx1-S pIX at nucleotide –4; Ad26.COV2.S pIX at nucleotide –5). In wildtype adenoviruses this site is naturally used as SA during processing of the adenoviral E1B RNA. To investigate potential splicing between the Spike sequence and the adenoviral pIX, HeLa and HepG2 cells were transduced with the adenoviral vectors Ad5.S, ChAdOx1-S, or Ad26.COV2.S. Preceding uptake assays for each cell line had been performed to determine required multiplicities of infection (MOIs) of the different adenoviral vectors to achieve comparable cellular transduction efficiencies with the different adenoviral vector types (Figure 8). Two days post transduction, total RNA was extracted from the cells and analyzed by RT-PCR using universal pIX reverse primer to detect splice events between various locations of the Spike-coding RNA and the pIX transcript. As shown in *Figures 5 and 6*, *Figure 7* all three Spike genes of the different adenoviral vector backbones displayed splice events to the adenoviral pIX transcript. Noteworthy, although all Spike genes contain a poly(A) signal sequence that should initiate the poly-adenylation process, in all cases we observed transcriptional read-through allowing the detected splice events to occur. Only recurrently identified splice events are displayed in *Figures 5–7*. The Ad5.S Spike gene displayed the predominant splice events at positions 506, 1821, 3610*, 3,614, and 3641* to the SA within the pIX transcript of the vector backbone (numbers with an asterisk were not predicted by SpliceRover). The ChAdOx1-S Spike gene displayed the predominant splice events at positions 506, 1049*, 1914*, 3610*, 3614 and –298* and –84*. The latter two splice events involved an SD located in the bovine growth hormone (BGH) poly(A) signal 298 and 84 nucleotides upstream of the pIX start codon. Finally, the Ad26.COV2.S Spike gene displayed weak splice events occurring at nucleotide 541*, 959*, 1184, 1221*, 3641*, and –83*. Also here one splice event involved an SD in the simian virus 40 (SV40) poly(A) signal sequence located –83 nucleotides upstream of the pIX start codon.

## Predominant and cell type-dependent prevalence of ChAdOx1-S Spike RNA splicing

To validate the previous analyses of observed splicing between the adenoviral vector-encoded Spike and adjacent adenoviral protein pIX transcripts, we performed additional PCR on cDNA samples of HeLa cells transduced with equal efficiencies (*Figures 8 and 9*) with Ad5.S, ChAdOx1-S, and Ad26.COV2.S vectors. Using forward primers that bind to the Spike sequence and reverse primers that bind to the pIX sequences revealed for all three vectors read-through events with DNA fragment lengths

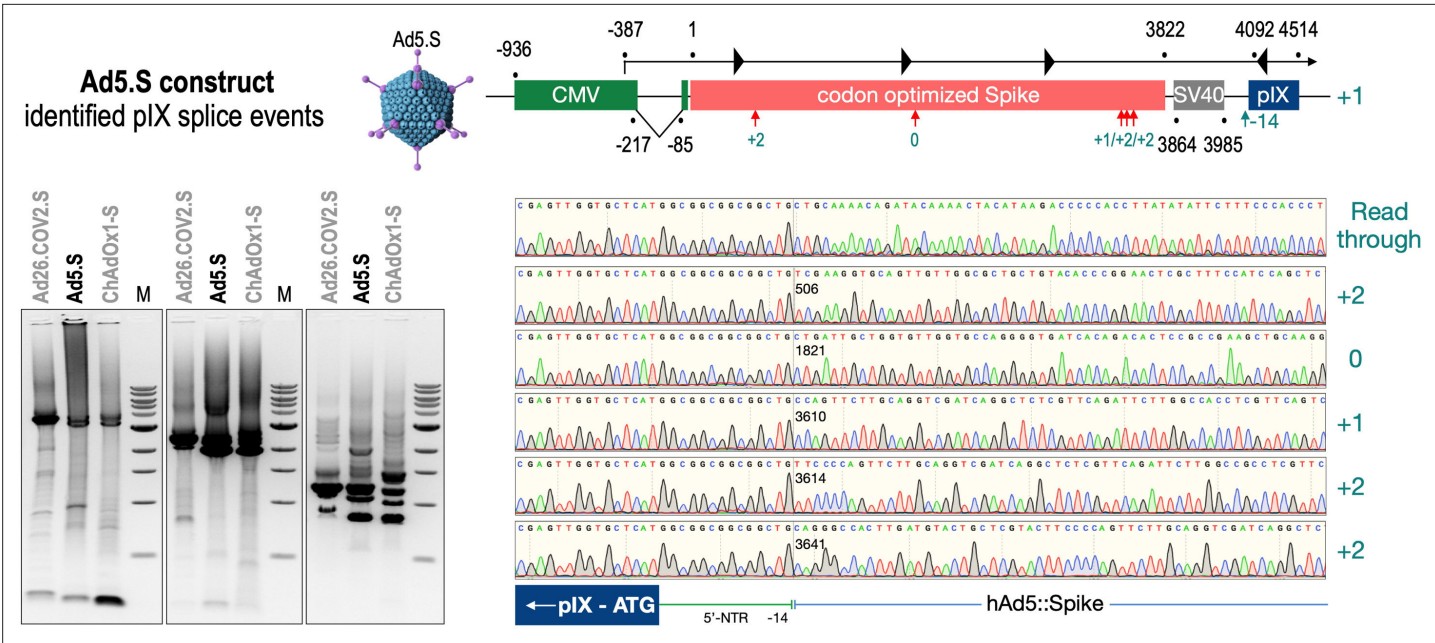

**Figure 5.** Splice events after Ad5.S transduction of HeLa cells. HeLa or HepG2 cells were transduced with the adenoviral vectors Ad5.S and RNA was isolated 48 hr post transduction. Three universal primers binding at three different locations within the Spike genes and one universal reverse primer binding to the adenoviral pIX that is located directly adjacent to the Spike gene sequence were used to investigate splicing events. The resulting DNA fragments are shown in the agarose gel on the left. Appropriate PCR bands were excised from the gel and either directly sequenced or cloned for sequencing. The predominately detected splice events are displayed on the right.

of 1802, 2037, and 1865 bp for Ad5.S, ChAdOx1-S, and Ad26.COV2.S, respectively (*Figure 9*, green arrows). While in Ad26.COV2.S-transduced HeLa cells no truncated cDNA fragments were detected, PCR amplification of Ad5.S- and ChAdOx1-S-transduced HeLa cells samples showed additional DNA bands of shorter lengths, confirming splicing at different positions within the read-through sequences. Interestingly, sequencing of the shortest DNA fragment of Ad5.S-transduced samples (*Figure 9*, red arrow) revealed that splicing occurred in-frame from the predicted SD site upstream of the TM domain of the Spike protein to the SA site upstream of the pIX start codon (*Figure 5*, splice site at nt 3610), resulting in a potentially secreted fusion protein. Similar results were obtained from sequencing of the shortest DNA fragment of ChAdOx1-S-transduced samples (*Figure 9A*: blue arrow): splicing took place between the predicted SD site upstream of the TM of the Spike protein (*Figure 6*, splice site at nt 3610) to the SA site upstream of the pIX start codon (*Figure 9F*). However, this splicing event resulted in the generation of a stop codon, thus no fusion but though secreted, truncated proteins would be translated due to the lacking TM domain. Sequencing of the most abundant DNA fragment in ChAdOx1-S samples (*Figure 9*, white arrow) revealed splicing from the predicted SD site within the BGH poly(A) tail to the SA site upstream of the pIX start codon (*Figure 6*, splice site at nt –298).

Comparing the total amount of PCR-generated DNA indicates significant higher Spike-pIX read-through copy numbers (and subsequent potential splicing) for Ad5.S-transduced HeLa samples than ChAdOx1-S-transduced HeLa samples (note even 2.5-fold less volume of PCR samples was loaded for Ad5.S; *Figure 9A*). Since expression cassettes of both vectors are under the control of the identical hCMV promoter, this data suggests that the BGH poly(A) of ChAdOx1-S is more potent than the SV40 poly(A) of Ad5.S to prevent read-through transcription into the pIX gene.

Interestingly, the use of cells of other tissue origin, in particular the human hepatocyte cell line HepG2 and the murine C2C12 myocytes, demonstrated that the abundance of splicing events has been cell type-dependent. In cDNA samples of Ad5.S-transduced C2C12 cells, only the full-length read-through fragment was detected – indicating no splicing at all (*Figure 9C*: green arrow). In contrast, for ChAdOx1-S-transduced C2C12 cells almost exclusively the BGH-pIX splice product was detected (*Figure 9C*: white arrow and *Figure 9E*). Moreover, in hepatocytes the ratio between full-length and spliced products was again different for Ad5.S- or ChAdOx1-S-transduced cells (*Figure 9B*). While PCR patterns of Ad5.S-transduced HepG2 cells looked the same as with HeLa cells,

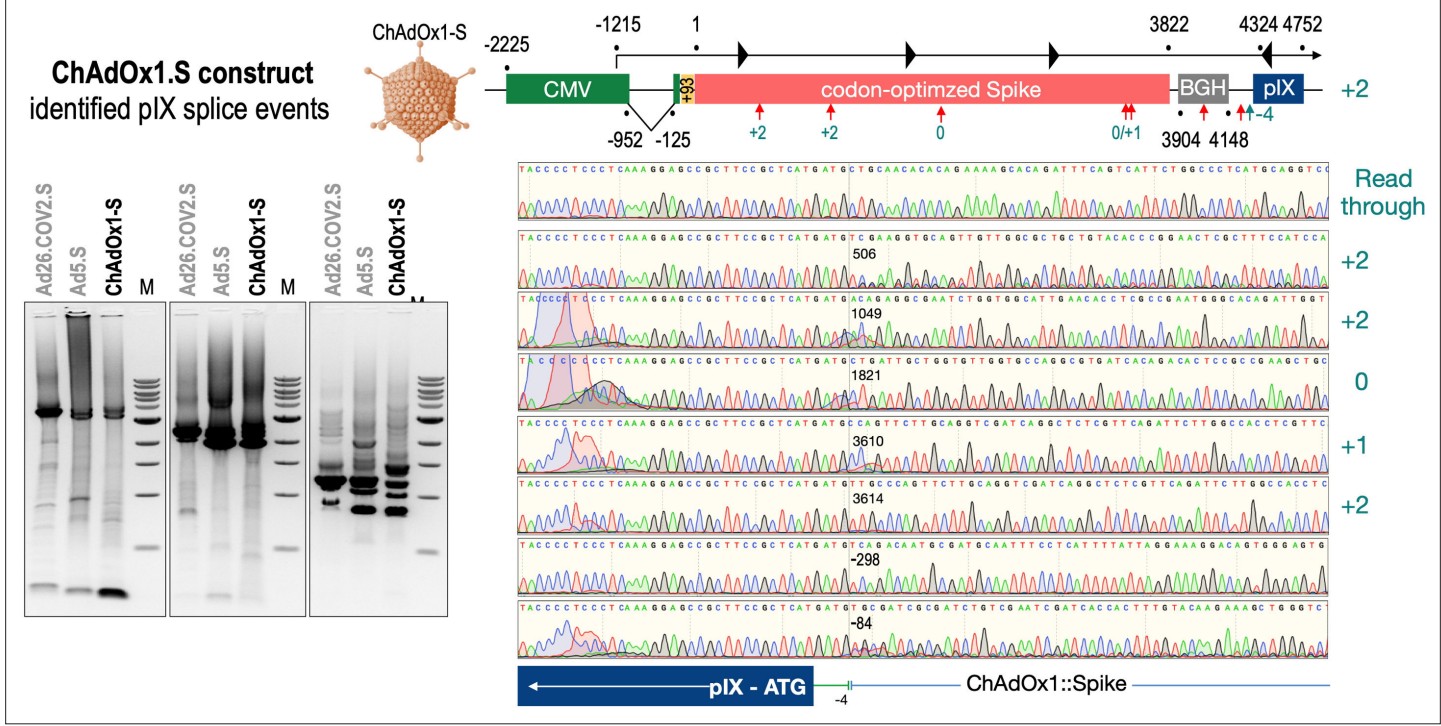

**Figure 6.** Splice events after ChAdOx1-S transduction of HeLa or HepG2 cells. HeLa or HepG2 cells were transduced with the adenoviral vector Ad5.S and RNA was isolated 48 hr post transduction. Three universal primers binding at three different locations within the Spike genes and one universal reverse primer binding to the adenoviral pIX that is located directly adjacent to the Spike gene sequence were used to investigate splicing events. The resulting DNA fragments are shown in the agarose gel on the left. Appropriate PCR bands were excised from the gel and either directly sequenced or cloned for sequencing. The predominately detected splice events are displayed on the right.

in ChAdOx1-S-transduced HepG2 cells the Spike-pIX splice product was the predominantly detected fragment (*Figure 9B*: blue arrow and *Figure 9F*). Since these splice products lack the TM-coding sequence, the resulting protein will likely be secreted.

## Significant amounts of splice events detected in genome-wide RNA-Seq data of ChAdOx1-transduced cells

To broaden our perspective regarding the occurrence of splicing upon cell transduction by ChAdOx1 from a pIX-focused to a genome-wide level, we also analyzed publicly available RNA-Seq data sets of ChAdOx1-S-transduced MRC-5 cells (*Almuqrin et al., 2021*). These data sets were based on nanopore sequencing data, which by nature are rather error-prone and thus analysis is complicated. Therefore, we first filtered the data sets for reads containing the ChAdOx1-S Spike gene. In a second step, we both looked for the presence of pIX sequences and blasted all Spike sequence reads against the human genome. The result of such an exemplary data mining experiment is shown in *Figure 10*, where the analysis of the data set derived from MRC-5 cells at day 72 after transduction with ChAdOx1-S is displayed. Out of millions of reads, 9632 reads could be attributed to the codon-optimized Spike gene. As schematically depicted in *Figure 10*, the Spike open reading frame is flanked at the 5'-end by the CMV promoter with a small intron and at the 3'-end by the BGH poly(A) signal and the pIX gene, respectively. A 93 nucleic acid sequence coding for the signal sequence of the tissue plasminogen activator (tPA) protein replaces the original start codon of the Spike gene. In this analysis, we identified, within the 9632 reads containing Spike nucleotide sequences, a total of 120 trans-spliced mRNAs, containing either host cell gene exons (fusing a total of 1 until 18 exons) fused to either the genuine SA of the CMV promoter (n = 8) or the internal Spike SA sites (n = 47). Similarly, we found Spike sequences fused to exons of host cell genes (between 1 and 28 exons). These trans-splicing events occurred either from the Spike open reading frame (n = 32, 32 different SD sites) or from the identified BGH SD sites (n = 33, 1 SD site). Exemplarily, we have identified the following genes: RSP, MSPG2, GPN2, CCN1, MRPL9, SORT1, UROS, TMEM35B, PRDX, ADAR, IFITM1, MMP1, DUD, PKM,

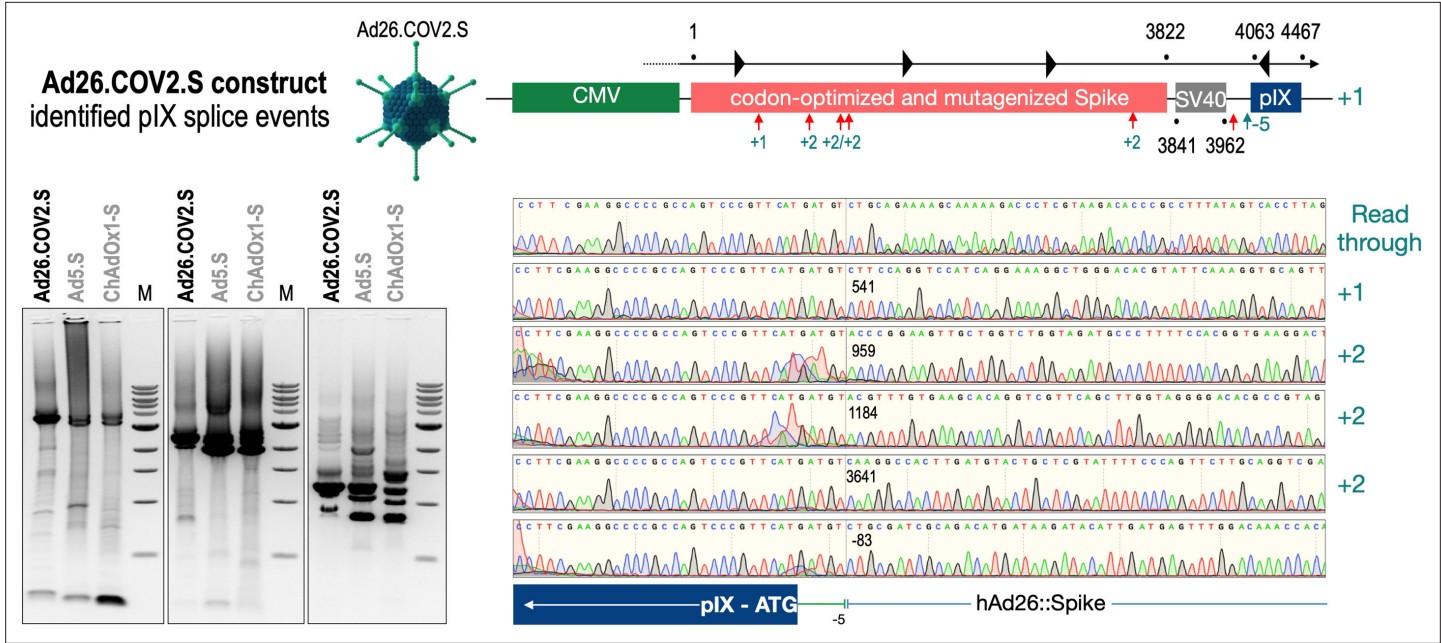

**Figure 7.** Splice events after Ad26.COV2.S transduction of HeLa or HepG2 cells. HeLa or HepG2 cells were transduced with the adenoviral vectors Ad5.S and RNA was isolated 48 hr post transduction. Three universal primers binding at three different locations within the Spike genes and one universal reverse primer binding to the adenoviral pIX that is located directly adjacent to the Spike gene sequence were used to investigate splicing events. The resulting DNA fragments are shown in the agarose gel on the left. Appropriate PCR bands were excised from the gel and either directly sequenced or cloned for sequencing. The predominately detected splice events are displayed on the right.

ALDOA, MT2A, DDX42, etc. A total of 50 Spike-pIX fusions were found as well (eight SD sites), while 79 reads were found from the BGH SD to the SA upstream of the pIX-coding sequence as it has been observed with HeLa, HepG2, and C2C12 cells. Although the frequency of these splicing event was rather low compared to normal Spike reads (1.24–1.33% vs. 97.41%), their identification was unexpected. For comparison, we also determined the number of reads for two housekeeping genes, ABL1 and GAPDH, respectively: ABL1 reads were only 44, while GAPDH reads were 4586. Thus, the amount of 249 reads that displayed aberrant forms of Spike should be considered as significant.

## Discussion

Here, we provide experimental evidence that adenovirus-based vaccines designed to express the SARS-CoV-2 Spike protein from a cloned, codon-optimized cDNA may result in unwanted splicing events, if the presence of functional splicing sites in the coding sequence is not avoided during vector design.

SARS-CoV-2 is an RNA virus with an infectious cycle that takes place entirely in the cytoplasm. Thus, there has never been an evolutionary selection against presence of SD and SA sites in its gene sequences. While mRNA-based vaccines that are functionally only active in the cytoplasm, transcripts from adenoviral vectors are generated in the cell nucleus, where also post-transcriptional RNA processing steps – including splicing – is taking place. Thus, the findings shown and discussed here have implications for DNA-encoded vaccine development not only against SARS-COV-2 but also against other RNA viruses.

In this study, we experimentally validated our hypothesis that Spike protein variants lacking the natural membrane anchor – thereby being subjected to secretion – are generated from adenoviral vaccine vectors as a consequence of nuclear RNA splicing events. We showed (1) that in a Spike/splice reporter system use of in silico predicted SD sites within the wildtype Spike gene resulted in Spike-Luciferase protein fusion events, (2) that in this system splice reactions lead to secreted protein variants due to the splicing-dependent loss of the TM membrane domain, (3) that two of the three evaluated adenoviral vectors that encode a Spike gene that is codon-modified for improved expression exhibited more and stronger SD sites than the viral wildtype sequence, and (4) that following

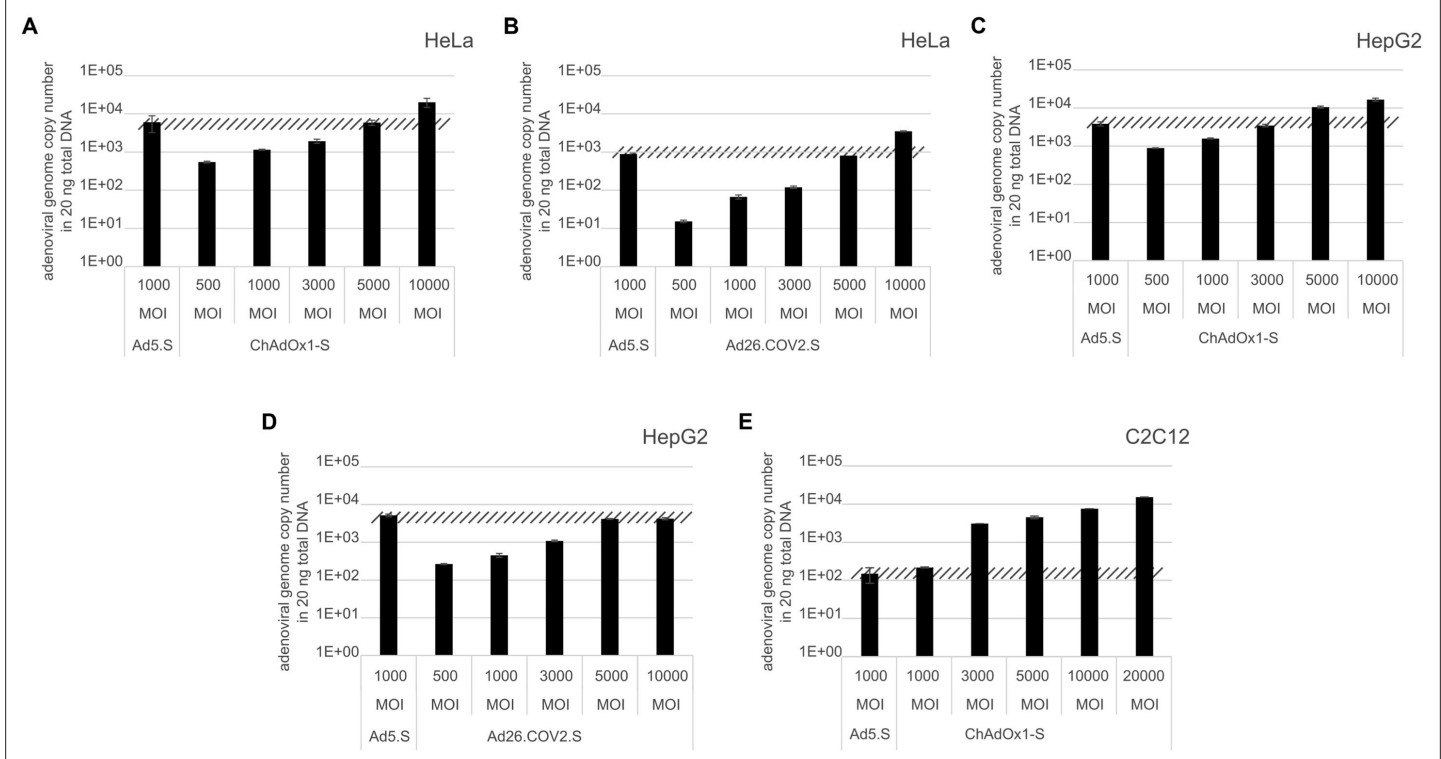

**Figure 8.** Analysis of required multiplicity of infection (MOI) of different vector strains to achieve equal transduction efficiencies. HeLa, HepG2, and C2C12 cells were transduced with indicated physical MOI of Ad5.S, ChAdOx1-S, or Ad26.COV2.S. Cells were incubated with the vectors for 2 hr at 37°C and harvested. Total DNA was isolated and intracellular adenoviral genome copy numbers were analyzed by quantitative real-time PCR.

transcriptional read-through, splicing occurred to a strong SA in the adenoviral pIX transcript, going also along with loss of the TM anchor of Spike.

We wish to point out that in our experimental studies we have investigated a distinct spectrum of splice events, since we used primers binding to either Luciferase or the adenoviral pIX-coding sequence for PCR amplification. Alternative splice events, for example, internal splice events (in-frame, out-of-frame) were mostly cloned and sequenced by chance. The analysis of provided RNA-Seq data after cells were infected with the AstraZeneca vaccine gave insight into additional trans-splicing events with host cell genes. Viral RNA genes evolutionary contain splice site consensus sequences due to a missing counter selection. Thus, we conclude that nuclear expression of such RNA virus-derived genes in mammalian cells may result in unwanted splicing events with host cell genes that should be considered when developing new vaccines in the future.

Our results clearly indicate that Spike splice reactions occur and that secreted soluble Spike protein variants are generated. Western blot analyses from total cells and supernatant thereof revealed a series of Spike protein variants that vary in their molecular weight, indicative for the previously analyzed splicing events. Soluble Spike protein has been described to cause adverse effects, for example, a strong inflammatory response on endothelial cells (*Lei et al., 2020*; *Nuovo et al., 2021*; *Patra et al., 2020*; *Amraei and Rahimi, 2020*). Obviously, severe life-threatening thromboembolic events occur due to infection of recipient cells by wildtype SARS-CoV-2. Even pseudoviruses with Spike protein located at their surface may cause strong inflammatory reactions in tissues and endothelial cells, indicating a potential risk of soluble Spike protein to cause severe side effects when systemically present in the vascular system (*Lei et al., 2021*).

According to our results, soluble Spike variants could potentially explain the occurrence of rare but severe events observed in particular after vaccination with Vaxzevria. Noteworthy, the Ad26.COV2.S vaccine carries fewer SD sequences, especially $SD_{506}$ and $SD_{3614}$ are absent (see *Table 2B–D*), which are the strongest predicted SD sites in the Ad5.S- and ChAdOx1-S-encoded Spike sequences (see *Figure 1*). This was further confirmed by the observation that in contrast to Ad5.S and ChAdOx1-S, no

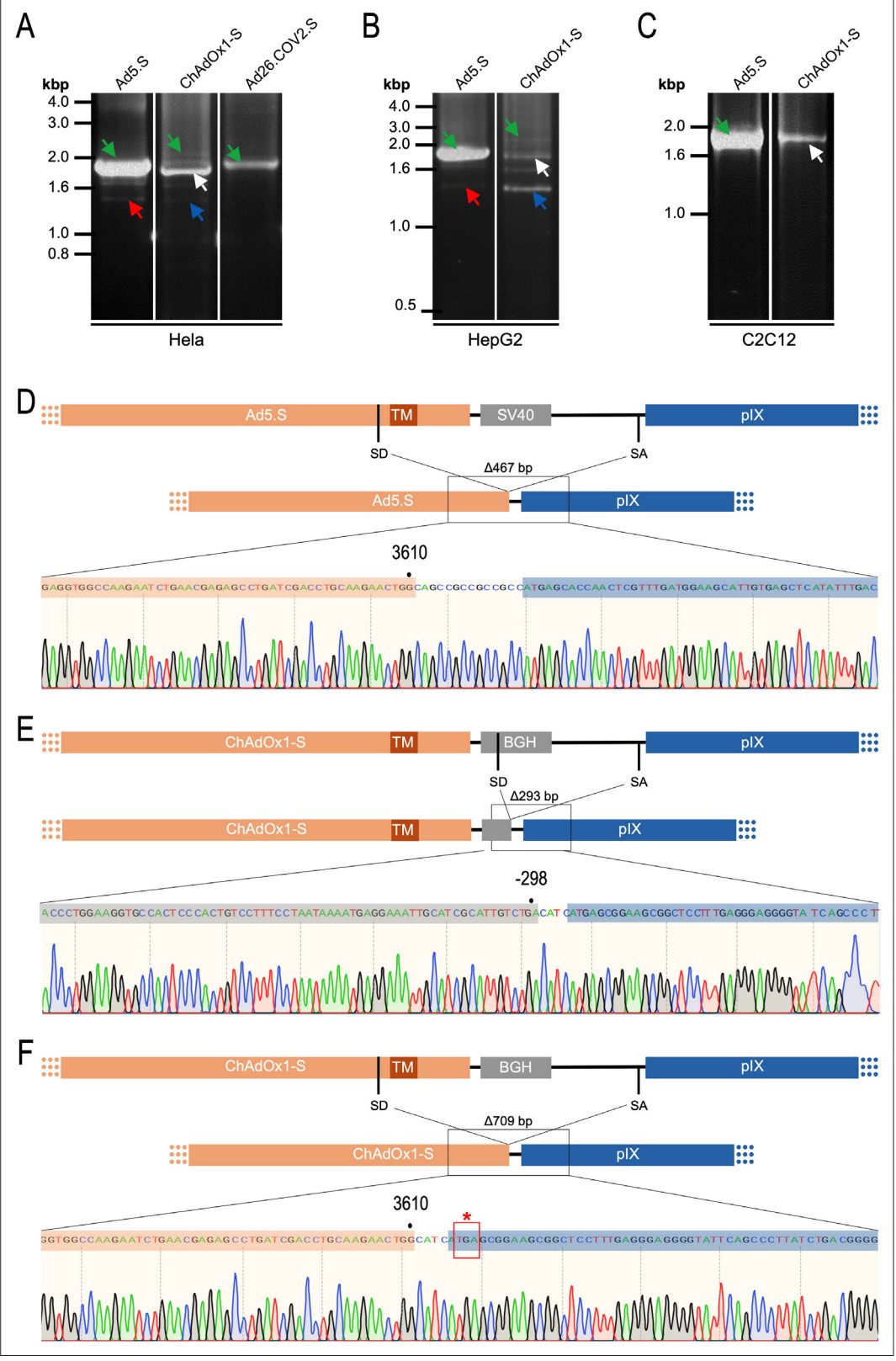

**Figure 9.** Cell type-dependent and predominant splicing of the ChAdOx1.S Spike sequence. (**A**) HeLa, (**B**) HepG2, and (**C**) C2C12 cells were transduced with equal efficiencies with the adenoviral vectors Ad5.S, ChAdOx1-S, and Ad26.COV2.S encoding the codon-optimized Spike sequences. Total RNA was extracted 48 hr post transduction. PCRs using forward primers that bind to the Spike sequence and reverse primers that bind to the adjacent

*Figure 9 continued on next page*

*Figure 9 continued*

downstream pIX in the adenoviral DNA genome were used for PCR amplification of generated cDNAs. Separation of PCR amplicons by agarose gel electrophoresis. Green arrow: Full-length read-through fragments. Red arrow: Spliced fragment of the Ad5.S-encoded Spike sequence resulting in a potentially secreted Spike fusion protein. White arrow: Spliced fragment of the ChAdOx1-S-encoded Spike sequence resulting in a Spike-pIX fusion protein. Blue arrow: Spliced fragment of the ChAdOx1-S-encoded Spike sequence resulting in a potentially secreted Spike protein. (**D**) Sequencing result of DNA fragment marked with the red arrow. (**E**) Sequencing result of DNA fragment marked with the white arrow. (**F**) Sequencing result of DNA fragment marked with the blue arrow.

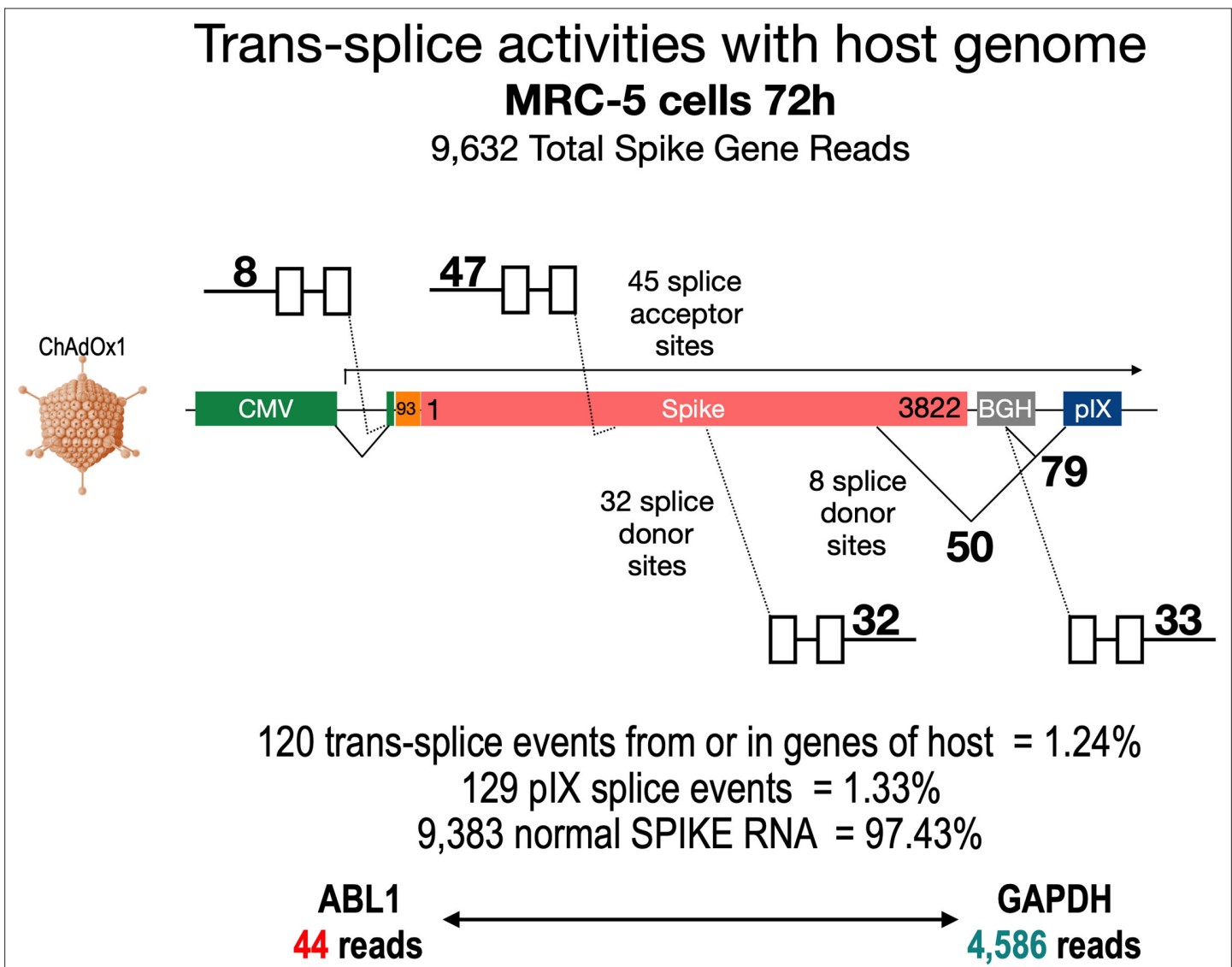

**Figure 10.** Trans-splicing events of Vaxzevria in MRC-5 cells. RNA-Seq data were kindly provided by David A Matthews, University of Bristol. RNA-Seq data derived from MRC-5 cells after transduction with the Vaxzevria vaccine. RNA was isolated after 72 hr and sequenced in depths. We used this data set to investigate potential splice events visible in this cell line. Total numbers and locations of detected trans-splice and pIX-splice events within the Spike sequence and the adjacent pIX sequence are graphically depicted. As displayed, out of 9632 Spike reads 120 reads were obtained from transsplicing events from or to host mRNAs; 129 reads were found that demonstrated splicing upstream of the pIX gene. We also determined the amount of two housekeeping genes (ABL1 and GAPDH) to analyze their read numbers. In comparison to the housekeeping genes, vector-based transcription was quite strong and superseded the amount of GAPDH transcripts.

significant amounts of spliced RNA molecules upon cell transduction with Ad26.COV2.S were generated. This may explain the ~3-fold lower incidence for thromboembolic side effects with the Janssen vaccine when compared to the Vaxzevria vaccine (see *Table 1*). It may also explain the ~2-fold lower risk for TTS with the Janssen vaccine.

Based on our data we pose the hypothesis that soluble Spike via its receptor-binding domain will bind to ACE2 and anti-Spike antibodies generated during immunization will recognize Spike variants, bound to the endothelial cell surface. Inflammatory reactions could locally occur through different mechanisms including antibody-dependent cell-mediated cytotoxicity (ADCC) and complement-dependent cytotoxicity (CDC), both of which could serve as starting point for thrombus formation. In ADCC, mainly NK cells are recruited to surface-bound antibodies via Fc-gamma receptors (CD16 or CD32) and cause cell lysis after their activation. In CDC, anti-Spike antibody-coated endothelial cells may recruit and activate complement that may result in platelet activation via binding of C3 and terminal complement complex on the surface of platelets and thrombin/serotonin secretion (*Luo et al., 2020*). By this means, thromboses may occur in any site of the human body where endothelial cells express ACE2.

Why should such events occur in the cerebral dural venous sinus system? It is conceivable that due to the absence of venous valves bi-directional and posture-dependent blood flow in the dural venous sinus, the residence time of soluble Spike protein is increased and the possibility to bind to endothelial

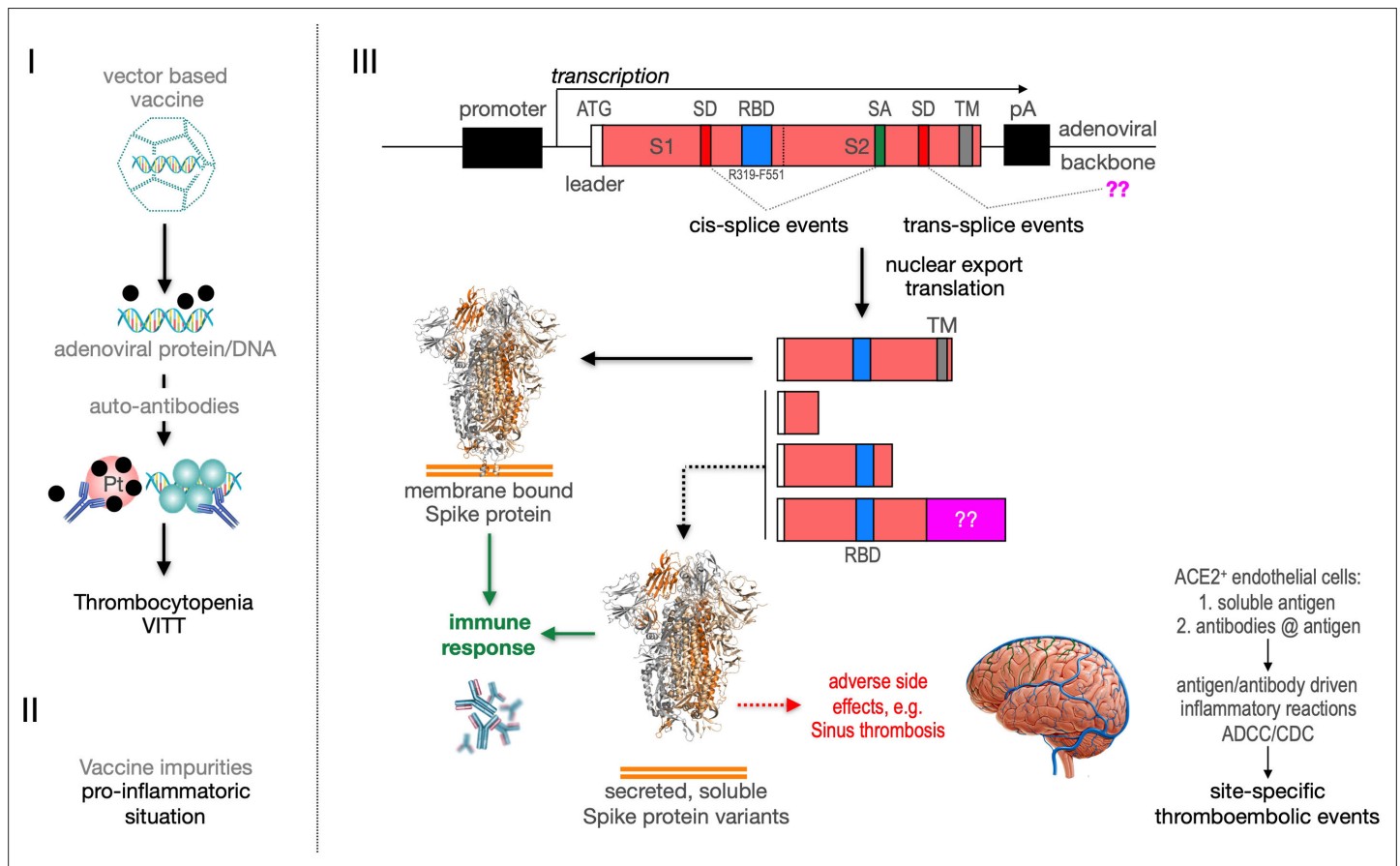

**Figure 11.** Predicted pathological disease mechanism.
 I. Predicts vaccine-induced immune thrombotic thrombocytopenia (VITT) mechanism that is accompanied by the thrombocytopenic situation. II. Pro-inflammatory events which might be caused by vaccine impurities. III. Proposed mechanism that soluble Spike protein variants are generated by distinct splice events. Soluble Spike protein variants which are able to bind to ACE2-expressing endothelial cells may trigger immunological mechanisms that result in thromboembolic events (cerebral venous sinus thromboses [CVST] and splanchnic vein thromboses [SVT]; estimated 1 in 80,000 vaccinated persons in Europe). RBD: receptor-binding domain; TM: transmembrane domain; SD: splice donor; SA: splice acceptor, pA: poly-A sequence;??: fusion to unknown sequences.

**Table 3.** Comparison of all three adenoviral vector-based constructs.

Based on our analysis of three different adenoviral vectors, there were differences between the vaccines. The displayed table analyzes the use of promoter, the different poly(A) elements, and changes inside the sequences of Spike, for example, additional sequences, presence or deletion of the furin cleavage sequence, the addition of two prolines to stabilize of the prefusion configuration of Spike, and site-directed mutations that should avoid splicing.

| Name | Vector | Promotor | Additional sequences | Terminator | RRAR | PP | Splicing |
|---|---|---|---|---|---|---|---|
| Ad5.S | HAd5 | CMV with Intron | – | SV40 | Present | No | + |
| ChAdOx1-S | ChAd | CMV with Intron | + tPA (31aa) | BGH | Present | No | + |
| Ad26.COV2.S | HAd26 | CMV w/o Intron | – | SV40 | Deleted | Present | Splice sites partially deleted |
| Comment | | | Upstream ATG | | pos 682–685 | pos 986/987 | AZ >> J |

cells expressing ACE2 enhanced. This could increase the probability of adverse immune reactions at places of slow blood flow.

Recently the VITT syndrome has been described, suggesting a complex inflammatory process. Related to HIT it has been reported that the induction of anti-PF4 autoantibodies and platelet activation in a subset of patients may result in the observed combination of thrombocytopenia and thromboses mainly in large vessels including cerebral venous sinus (*Figure 11*, left side). Additionally, it is possible that protein impurities from the vaccine production process may be involved in the generation of a pro-inflammatory status (*Krutzke et al., 2021*). Here, we propose a third pathomechanism as it is depicted in *Figure 1*, right side. Based on our splicing data, membrane-anchored and secreted soluble Spike protein variants are produced after vaccination. Secreted Spike protein variants are disseminated throughout the vascular system and may concentrate at endothelial cells expressing the ACE2 at their surface. These ACE2-bound Spike will then become a target of anti-Spike antibodies, generated after vaccination with the potential to result in ADCC- and/or CDC-mediated inflammatory reactions. It may also cause the already described NET reaction of neutrophils that has been causally linked to thromboembolic events (*Schönrich et al., 2020*; *Veras et al., 2020*; *Middleton et al., 2020*). Also the rarely observed capillary leak syndrome (Clarkson disease; *Kawabe et al., 2002*) could potentially be explained by such a mechanism.

The distinguishing features of the three different adenovirus vectors Ad5.S, ChAdOx1-S, and Ad26.COV2.S have been summarized in *Table 3*. Based on the analysis of the design of the vaccines and their splice behavior of the encoded transcripts as described in this report, we have identified several aspects that should be considered for future adenovirus vaccine design and development.

We have two different data sets. One data set has been created by large-scale sequencing of splice events of all three adenoviral vector systems after transduction in HeLa and HepG2 cell lines (*Figures 5–7*), where the different splice events of all three vectors have been investigated by simply analyzing a large number of PCR fragments through direct sequencing or subcloning and sequencing. The second data set is based on experiments, where all three adenoviral vectors were analyzed for their splice activity after adjusting the MOIs in the HeLa, HepG2, and C2C12 cell lines (*Figure 8*). In the latter, read-through transcripts have been investigated for the activity of the different poly(A) signal sequences (BGH vs. SV40), but also for the amount of splice events in that particular transcript (*Figure 9*). Transcriptional read-through obviously is a precondition for the generation of splice events between the Spike-coding sequence and the pIX gene encoded by the adenoviral vector genome. Very little transcriptional read-through was observed with ChAdOx1-S that can be linked to the BGH poly(A) signal used in this construct. Apparently, the SV40 poly(A) signal is not as efficient in termination of the transcript as we observed with Ad5.S. In these cases, differences were observed for splice events which may point to the fact that tissue-specific splicing may account as well as for some differences in the creation of soluble Spike protein variants.

Based on our findings, we suggest that the Spike open reading frames – wildtype or codon-optimized for expression – in DNA-based vaccines should always be analyzed for the presence of strong SD sites. If possible, strong SD should be avoided to prevent unintended splice. We note that the Janssen Ad26.COV2.S vaccine has been modified to reduce the number of strong SDs, while the Ad5.S and ChAdOx1-S vectors still contain strong SDs, which is reflected in a large number of

detectable splicing events, while only very few splicing events were detected in the Ad26.COV2.S vaccine.

Very little is known about fate and tissue distribution of adenovirus-based vaccine in humans. In mice, it has been observed that following intramuscular injection significant amounts of adenovirus vectors end up in hepatocytes, where expression of the transgene occurs (*Kron et al., 2011*). It is likely that the chimpanzee ChAdOx1-S is also a hepatotropic adenovirus, since it is based on the chimpanzee adenovirus Y25 (*Dicks et al., 2012*), which belonged to a group of isolates from chimpanzees associated with outbreaks of viral hepatitis (*Hillis and Goodman, 1969*). If transduction of hepatocytes would occur in (some of the) vaccines after intramuscular injection in a significant amount, it is conceivable that larger amounts of Spike proteins, having lost the transmembrane domain by splicing, will be secreted into the systemic circulation via the hepatic veins.

Taken together, from our work it appears that for DNA-encoded sequences (especially when derived from RNA viruses), most importantly, strong SDs should be avoided and strong poly(A) signals should be used for RNA termination in expression constructs. This is especially important when adenoviral vectors are used as transport vehicles. Usually, transgenes sequences are embedded in the adenoviral genome in the gap of the deleted *E1*-region, located directly upstream of the adenoviral pIX with a strong SA site. Alternatively, while most transgenes in adenoviral vectors have a 'left-to-right' orientation, also an inversion of the expression cassette would avoid splicing events to occur between the transgenic-coding sequence and the viral pIX transcript. Taking this into account, chances of splicing will be strongly reduced or null.

We strongly emphasize the consideration of these aspects for the rational design of safer future adenoviral vaccine vectors and also for adenoviral vector-based gene transfer approaches in general.

## Materials and methods
### Splice site prediction
The SARS-CoV-2 Spike nucleic acid sequences of of the Wuhan SARS-CoV-2 strain (NC_045512.2) and codon-optimized Spike sequences encoded by the three investigated adenoviral vectors Ad5.S, ChAdOx1-S, and Ad26.COV2.S were analyzed for the presence of SD and SA sites using the state-of-the-art SpliceRover online tool (bioit2.irc.ugent.be/rover/splicerover). Sites with a score below 0.15 were excluded from further analysis (*Zuallaert et al., 2018*). The ASSP online tool was used as second algorithm for potential splice site identification (*Wang and Marín, 2006*).

Cloning of the open reading frame of SARS-CoV-2 Spike protein cDNA prepared from the RNA of the Wuhan SARS-CoV-2 strain (NC_045512.2) was kindly provided by Tuna Toptan and Marek Widera (Institute of Medical Virology, University Hospital Frankfurt am Main, Goethe-University, Frankfurt am Main, Germany). The cDNA was used for PCR experiments and for cloning of the open reading frame of the Spike gene following amplification using the following primers: CoV-2_S.Flag.Sfi.F 5'-aGGCC TCTGAGGCCaccatggattacaaggatgacgacgataagatgtttgtttttcttgtttattgccactagtctct-3', CoV-2_S.Sfi.R 5'-aGGCCTGACAGGCCttatgtgtaatgtaatttgactcctttgagcac-3'. Resulting amplimers were cloned into the Topo II vector system and appropriate clones were validated by sequence analyses. The final clone was digested with Sfi1 and cloned into the Sleeping Beauty transposon vector system pSBTet-GP (GFP/Puromycin; *Kowarz et al., 2015*). The recombinant Spike was N-terminally modified with a FLAG-tag.

### Development of a Spike/splice reporter system
The cloned wildtype Spike open reading frame was used to establish an additional splice reporter system. The Spike/splice reporter system consists of the full-length Spike gene including the stop codon, followed by an intronic sequence that contains the necessary branch-A nucleotide, as well as a perfect SA site fused in three different reading frames (0, +1, +2) to a full-length Luciferase gene (without start codon). In this system Luciferase activity is only detected if splicing occurs. Thus, three independent Spike/splice reporter constructs (pSBbi::Spike-Luc-0/+1/+2 GP) were cloned into the integrating Sleeping Beauty transposon vector pSBbi-GP (*Kowarz et al., 2015*). Additionally, isolated adenoviral DNAs of Ad5.S, ChAdOx1-S, and Ad26.COV2.S were used as templates to PCR-amplify the 3822 bp codon-optimized open reading frames of the Spike sequences. These PCR fragments were also cloned into the three Luciferase reporter gene constructs pSBbi::LUC-0/+1/+2 GP.

The resulting Sleeping Beauty transposon vectors were sequence-validated and subsequently stably integrated into the genome of HEK293T cells for RT-PCR and Luciferase reporter experiments. The ChAdOx1-S Spike sequence constructs were cloned without the tPA leader sequence (+31aa) in order to make all three different constructs (Ad5.S, ChAdOX1-S, Ad26.COV2.S) equal in length. The AGA codon at position 94–96 of the original AZ vector was thus converted into an ATG start codon. This way, all tested construct had an open reading frame of 3822 nucleotides.

## Stable transfection experiments

All splice reporter constructs were stably transfected into HEK293T cells and selected for 3–5 days with 2 µg/ml Puromycin. Cells were used for Luciferase experiments or for the isolation of total RNA (Qiagen) in order to investigate potential splice events by RT-PCR.

## Adenoviral vectors

Ad5.S particles produced in N52.E6 cells (*Kowarz et al., 2015*) had a physical titer of $3.35 \times 10^8$ VP/µl and were dissolved in 50 mM HEPES, 150 mM NaCl, 10% glycerol, pH 7.4. According to the manufacturer, the ChAdOx1-S vaccine from AstraZeneca (lot number ABV9317) had a physical titer of $1 \times 10^8$ VP/µl and particles were dissolved in 10 mM histidine, 7.5% sucrose (w/v), 35 mM NaCl, 1 mM $MgCl_2$, 0.1% polysorbate 80 (w/v), 0.1 mM EDTA, and 0.5% EtOH (w/v). According to the manufacturer, Ad26.COV2.S vaccine from Janssen (lot number XD955 and 21C10-01) had a titer of $1 \times 10^8$ VP/µl and particles were dissolved in citric acid monohydrate (0.14 mg), trisodium citrate dihydrate (2.02 mg), ethanol (2.04 mg), 2-hydroxypropyl-β-cyclodextrin (25.50 mg), polysorbate-80 (0.16 mg), sodium chloride (2.19 mg).

## Production and purification of Ad5.S vectors

Ad5.S vector particles used in this study are *E1*- and *E3*-deleted replication-incompetent vector particles based on human adenovirus species C type 5 (based on GenBank AY339865.1, Δ441–3,522, and Δ28123–30813). The CMV promoter-controlled SARS-CoV2 Spike expression cassette with a SV40 poly(A) signal is based on the GenBank sequence YP_009724390.1, codon-optimized for *Homo sapiens* and was inserted in the *E1* region. Particles were produced in *E1*-complementing N52.E6 cells (*Schiedner et al., 2000*) and purified by CsCl gradient ultracentrifugation as described earlier (*Krutzke et al., 2020*). In brief: $4 \times 10^8$ cells were transduced with MOI 300 from stock solution; 48 hr post transduction cells were harvested, resuspended in 3 ml 150 mM NaCl, 50 mM HEPES, pH 7.4, and lysed by freeze/thaw cycles. Cell debris was separated by centrifugation at 2000× *g* for 10 min, and supernatants were layered on a CsCl step gradient (density bottom: 1.41 g/ml; density top: 1.27 g/ml, 50 mM HEPES, 150 mM NaCl, pH 7.4) and centrifuged at 176,000× *g* for 2 hr at 4°C. Vector particles were aspirated and further purified by a consecutive continuous CsCl gradient (density: 1.34 g/ml, 50 mM HEPES, 150 mM NaCl, pH 7.4) and centrifuged at 176,000× *g* for 20 hr at 4°C. Vector particles were aspirated and desalted by size exclusion chromatography using PD10 columns (GE Healthcare). Physical vector titers were determined by optical density measurement at 260 nm (*Mittereder et al., 1996*).

## Isolation of adenoviral DNA from vector stock solution

Adenoviral DNA was isolated from 200 µl vector stock solutions of Ad5.S, ChAdOx1-S, and Ad26. COV2.S using the QIAamp DNA Mini Kit (250). Eluted DNA was precipitated with EtOH and pellets resuspended in 10 µl 10 mM Tris buffer, pH 8.0.

## Cell lines

HEK293T cells were maintained in DMEM Low Glucose medium mixed with 10% (v/v) FBS, and supplemented with 2 mM L-glutamine, 100 U/ml penicillin, and 100 µg/ml streptomycin. HeLa (human cervical cancer), HepG2 (human hepatocyte carcinoma), and C2C12 (murine muscle myoblast) were cultivated in MEM, alphaMEM, or DMEM medium, respectively, all of which were supplemented with 10% FBS and 1% antibiotics. Cells were grown at 37°C in 5% $CO_2$ and a relative humidity of 95%. Cells were passaged twice a week except for C2C12, which were passaged three times a week. For HeLa and HepG2 cells adenoviral vector transduction was done 24 hr after cell seeding, whereas C2C12 cells were incubated for 72 hr after seeding to let them grow dense and start differentiation into

myocytes. Cell lines were obtained from ATCC (HEK293T) or DMSZ (HeLa, HepG2, C2C12) and are tested for their cell identity by STR profiling as provided by company. In addition, these cell stocks were tested for mycoplasma infections (as not already done by the company), before being expanded in cell culture for 14 days and subsequently frozen as cell stocks in liquid nitrogen. For all cell culture experiment, such cell stocks were thawed and used in cell culture for a maximum of several weeks.

## Cellular uptake assays with different adenoviral vectors

$1 \times 10^5$ HeLa, HepG2, and C2C12 cells were transduced with indicated physical MOI of Ad5.S, ChAdOx1-S, or Ad26.COV2.S. Cells were incubated with the vectors for 2 hr at 37°C before cells were harvested and total DNA was isolated using GenElute Mammalian Genomic DNA Miniprep Kit (Sigma) according to the manufacturer's instructions. DNA concentrations were determined by optical density measurement at 260 nm. Intracellular adenoviral genome copy numbers in 20 ng total DNA were analyzed by quantitative real-time PCR. Primers used for Ad5.S and ChAdOx1-S samples: fw.: 5'-TAGACGATCCCTACTGTACG-3'; rv.: 5'-GGAAATATGACTACGTCCGG-3'. Primers used for Ad5.S and Ad26.COV2.S samples: fw.: 5'-CAGGACGCCTCGGAGTACCTGAG-3'; rv.: 5' GGGGCCACCGT-GGGGTT-3'. DNA was dissolved in 2 µl and mixed with 10 µl SYBR Green (Kapa Biosystems), 0.4 µl 10 pmol/µl of each forward and reverse primer in a total volume of 20 µl. Thermocycles: first cycle: 10 min at 95°C; 40 cycles: 30 s 95°C, 30 s 60°C, 8 s 72°C; followed by a final cycle of 10 min at 72°C. Assays were performed in biological and technical triplicates.

## Analysis of SARS-CoV-2 Spike RNA splicing after adenovirus vector-mediated cell transduction

Adenoviral cell transduction assays were performed using HeLa, HepG2, and C2C12 cells. To achieve equivalent cell transduction efficiencies with adenovirus vectors of different strains, HeLa cells were transduced with MOI 1000 for Ad5.S, MOI 3000 ChAdOx1-S, and MOI 6000 for Ad26. COV2-S. HepG2 cells were transduced with MOI 1000 for Ad5.S, MOI 3000 for ChAdOx1-S, and MOI 10,000 for Ad26.COV2.S. C2C12 cells were transduced with MOI 1000 for both Ad5.S and ChAdOx1-S (see *Figure 8*). Cells were incubated with the vectors for 48 hr at 37°C, before RNA was isolated using RNeasy Plus Mini Kit (Qiagen) followed by cDNA generation using Maxima H Minus cDNA Synthesis (ThermoScientific) according to the manufacturer's instructions. cDNA was used for RT-PCR (see section RT-PCR experiments) or 2 µl cDNA was used for Taq polymerase PCR in a total volume of 25 µl. Forward primer binding to the Spike sequence: 5'-CAAGGACTTCGGCGGCTTCAA-3', used for all three vectors. Reverse primer sequences binding to the adenoviral pIX sequence: 5'-CCCATCACATTCTGACGCAC-3' for Ad5.S and ChAdOx1-S; 5'-TGCTGTCGAGCGACGAGTTC-3' for Ad26.COV2.S. Thermocycles: 3 min at 95°C; 40 cycles: 30 s 95°C, 30 s 55°C, 2 min 68°C; 8 µl of Ad5.S and 20 µl of ChAdOx1-S and Ad26.COV2.S PCR products were separated on agarose gels, DNA bands were excised, purified with gel extraction kit (Qiagen), cloned using the pCR 2.1-TOPO TA cloning kit (ThermoScientific) and sequenced using primers 5'-CAGGAAACAGCTATGAC-3' and 5'-GTAAAACGACGGCCAG-3'. Assays were performed in biological triplicates.

## RT-PCR experiments

For RT-PCR experiments a series of primers were used. To test for splice events in the three reporter cell lines, we used the following primer sequences: S.0233.F 5'-GGTTTGATAACCCTGTCCTACCA-3', S.1027.F 5'-AACGCCACCAGATTTGCATC-3', S.1712.F 5'-ACACTACTGATGCTGTCCGTG-3', S.2482.F 5'-CTTGCAGATGCTGGCTTCAT-3', and the reverse primer Luc4008.R 5'-GTCCACCTCGAT ATGTGCGTC-3'. RT-PCR experiments were carried out under stringent conditions, and PCR fragments deviating from expected PCR bands were analyzed by Sanger sequencing.

For the analysis of RNA derived from adenoviral vector transduced HeLa and HepG2 cells, we developed a series of universal primers. These were the following primers: 506univ.F 5'-GCGAGTTCCAGT TCTGCAACG-3', 1,810univ.F 5'-CAGACACTGGAAATCCTGGACATCAC-3', 3610 .F 5'-GCGCCAT-CAGCTCTGTGCTG-3', and the reverse primer pIXuniv.R 5'-CGTCAGAATGTGATGGGATCGACG-3'. RT-PCR experiments were carried out under stringent conditions, and PCR fragments deviating from expected PCR bands were analyzed by Sanger sequencing. Thermocycles: first cycle: 2 min at 94°C; 35 cycles: 20 s 95°C, 30 s 60°C, 40 s 72°C; followed by a final cycle of 3 min at 72°C.

## Luciferase assays

Luciferase assays were performed using HEK293T cells that were stably transfected with the Sleeping Beauty Luciferase reporter gene constructs that exhibit a constitutive GFP expression cassette; 5000 cells per well were seeded in black glass bottom 96-well plates (Greiner BioOne). After 2 days, GFP fluorescence (RFU) was measured with the Varioskan Flash Plate-reader (Thermo Fisher Scientific). Subsequently, D-luciferin (final concentration of 187.5 µg/ml) was added with the automatic dispense system, and Luciferase activity was determined (RLU). A normalization procedure was used to express the RLU/RFU ratio to account for intercellular differences. All measurements were carried out with n = 6, while Luciferase activity in medium was measured once at two different time points (t = 0 and t = 48 hr). For this purpose, 200 µl medium was centrifugated for 5 min at 800× $g$ and 100 µl supernatant was used to determine Luciferase activity.

## Immunoprecipitation and Western blot analyses

Cells were lysed using immunoprecipitation buffer (150 mM NaCl, 10 mM Tris, 1 mM EDTA, 1 mM EGTA, 1% Triton-X-100, 0.5% NP-40, protease inhibitor cocktail [Roche Diagnostics]) for 30 min under constant agitation at 4°C. Cell lysates were separated from cell debris by centrifugation for 5 min at 2000 rpm (376× $g$) and 4°C. Lysates were quantified using Pierce BCA Protein Assay Kit (Thermo Scientific) according to the manufacturer's protocol and adjusted to 40 µg/µl.

The amount of 200 µl of cell lysate or cell culture supernatant (3 days) was pre-cleared with 25 µl protein G magnetic beads (New England Biolabs) in low-binding tubes (Biozym Scientific) for 1 hr under constant agitation at 4°C. After magnetic separation the supernatant was incubated with 25 µl anti-FLAG M2 magnetic beads (Sigma-Aldrich #M8823) for 2 hr under constant agitation at 4°C. Beads were then washed three times with 0.5 ml IP buffer and resuspended in 20 µl 2× Läemmli buffer. For elution the precipitate beads were incubated for 5 min at 70°C and beads magnetically separated.

Ten µl of supernatant was electrophoretically separated using Mini-PROTEAN TGX stain-free gradient gels 4–15% (Biorad) and color protein standard broad range P7719S (NEB). Proteins were blotted onto low fluorescence western PVDF membrane (Abcam) in Towbin buffer (25 mM Tris, 192 mM Glycin, 15% methanol, 0.01% SDS, pH = 8.3) at 110 V for 70 min. Membranes were blocked using 5% BSA in TBS-T for 1 hr at room temperature. Primary antibody incubation was performed over night at 4°C. Membranes were washed with 5% BSA in TBS-T four times prior to incubation with HRP-conjugated secondary antibody for 1 hr at room temperature. After four wash steps with 5% BSA in TBS-T chemiluminescence detection was performed using Clarity Western ECL Substrate (Biorad) and ChemiDoc XRS + system (Biorad).

The following antibodies were used: α-Firefly Luciferase pAb (Abcam #ab21176) diluted 1/1000, α-SARS.CoV.2 Spike pAb (Sino Biologicals #40589-T62) diluted/1000, α-rabbit IgG-HRP (Abcam #ab6721) diluted 1/10,000.

## Acknowledgements

We thank all lab members for fruitful discussions and their help. We also want to thank Prof. Owen Williams for critical reading of the manuscript. We also thank Dr Tuna Toptan and Dr Marek Widera for providing us with the cDNA of SARS-CoV-2 (Wuhan isolate). We are also grateful to Dr Roland Zahn (Janssen) for providing the Spike gene sequence of the Janssen vaccine. This work was supported by a funding grant of the Corona Task Force of the Goethe-University to RM.

## Additional information

### Funding

| Funder | Grant reference number | Author |
| --- | --- | --- |
| Goethe University Corona Task Force | | Rolf Marschalek |

The funders had no role in study design, data collection and interpretation, or the decision to submit the work for publication.

## Author contributions

Eric Kowarz, Conceptualization, Data curation, Formal analysis, Investigation, Supervision, Writing - review and editing; Lea Krutzke, Conceptualization, Data curation, Formal analysis, Methodology, Supervision, Writing - review and editing; Marius Külp, Patrick Streb, Data curation, Formal analysis, Investigation; Patrizia Larghero, Data curation, Investigation, Software, Visualization; Jennifer Reis, Silvia Bracharz, Tatjana Engler, Formal analysis, Investigation; Stefan Kochanek, Conceptualization, Formal analysis, Methodology, Supervision, Writing - review and editing; Rolf Marschalek, Conceptualization, Funding acquisition, Project administration, Supervision, Visualization, Writing - original draft, Writing - review and editing

## Author ORCIDs

Lea Krutzke http://orcid.org/0000-0002-4092-4131
Rolf Marschalek http://orcid.org/0000-0003-4870-3445

## Decision letter and Author response

Decision letter https://doi.org/10.7554/eLife.74974.sa1
Author response https://doi.org/10.7554/eLife.74974.sa2

# Additional files

## Supplementary files

- Transparent reporting form
- Source data 1. Source original data 1.

## Data availability

The original WUHAN SARS-CoV-2 sequence is available in the NCBI database (NCBI Reference Sequence: NC_045512.2); the adenoviral and codon-optimized Spike sequence data have a protected intellectual property by the companies. The primary sequence of Ad5.S, designed and used by the colleagues in Ulm, can be retrieved upon request (contact Prof. Stefan Kochanek).

The following previously published dataset was used:

| Author(s) | Year | Dataset title | Dataset URL | Database and Identifier |
|---|---|---|---|---|
| Wu F, Zhao S, Yu B, Chen YM, Wang W, Song ZG, Hu Y, Tao ZW, Tian JH, Pei YY, Yuan ML, Zhang YL, Dai FH, Liu Y, Wang QM, Zheng JJ, Xu L, Holmes EC, Zhang YZ | 2020 | A new coronavirus associated with human respiratory disease in China | https://www.ncbi.nlm.nih.gov/nuccore/1798174254 | NCBI GenBank, NC_045512.2 |

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
