## [Editor Report]

In this manuscript, the authors provide evidence for the occurrence of splice reactions in adenovirus-based current vaccines resulting in the secretion of truncated Spike variants providing a potential mechanisms underlying thromboembolic events that have been reported for DNA-based but not for RNA-based COVID-19 vaccines.

---

## [Decision Letter]

**Decision letter after peer review:**

Thank you for submitting your article ""Vaccine-Induced Covid-19 Mimicry" Syndrome: Splice reactions in adenovirus-based current vaccines result in secretion of truncated Spike variants that may cause adverse side effects" for consideration by *eLife*. Your article has been reviewed by 2 peer reviewers, and the evaluation has been overseen by a Reviewing Editor and a Senior Editor. The following individual involved in review of your submission has agreed to reveal their identity: Hugo ten Cate (Reviewer #2).

The editors have discussed the reviews, and the Reviewing Editor has drafted this letter to help you prepare a revised submission.

Essential revisions:

1) Please revise the discussion to address the comments about the protein studies (see reviewer 1 and 2) and add what further data would help to further substantiate the hypotheses presented in the manuscript.

2) Reviewer 2: "Their data provide evidence that splicing of RNA originating from vector DNA is possible and might lead to alternative spike proteins. There is no evidence presented demonstrating a direct relation between the alternative extracellular proteins and pathogenicity either in in vitro or an animal model. Thus, the claim on page 10 lines 380-381 "According to our results, soluble Spike variants may be an additional causative factor for the occurrence of rare but severe events observed after vaccination with Vaxzevria" isn't supported by the presented data. And why is this statement made for Vaxzevria only and not for the Janssen vaccine? Is there a different mechanism for the latter?" Please revise according to this comment.

3) Reviewer 1: "line 45-46: the authors should be very careful with the interpretation of VITT after mRNA vaccines. Detailed analyses of the clinical presentation of these patients indicates rather coincidence than causality." Please consider adding this.

4) Reviewer 1: "line 17: the numbers given are much too low, the UK has documented more than 300 cases, Germany has documented more than 150 cases. Either the authors should omit the numbers or provide a reference for the data are coming from." Please update the data and add a reference

5) Reviewer 2: Page 2, lines 72-74: "The most severe side effects entailed rare events of thrombocytopenia combined with cerebral venous sinus thrombosis (CVST) causing the patients' death in about one third of the cases." This statement might need some clarification as it is not clear if it is one third of the 150 cases, of the 6 cases or the overall 156 cases. Also, one third seems a lot but giving a percentage based on the >24 million doses given will change the picture dramatically." Please update the manuscript with data from recently published reference of the absolute risks, such as DOI: 10.1212/WNL.0000000000013148

6) Please add references where missing for specific claims, as suggested by both reviewers.

7) Reviewer 1: "General format of all figures: figure legends are so small that they are already difficult to read in the provided figures, but will be impossible to read if the figures are printed in smaller size." Please improve the figures to improve readability.

*Reviewer #1:*

This study investigates the transcription mechanism and resulting spike protein production of adenoviral vector DNA encoding the SARS CoV2 spike protein. The main finding is that the transcribed hnRNA is processed by the splice machinery of the human cell. This results in truncated constructs of the spike protein, which lose the transmembrane part and can be secreted into circulation. Splicing, however, is highly dependent on the cell type. Production of spliced spike proteins have been shown in the study using HEK cells, but the controls in this experiment a week. No evidence is given that human muscle cells can produce spliced spike protein. The authors show that a murine myoblast cell line can splice spike protein mRNA, but no data are given whether the murine cell line also produces and secretes the spliced protein. The authors speculate that these truncated forms of the spike protein bind to human endothelial cells and induce downstream biological effects. Taken together, the mRNA studies are interesting and hypothesis generating, the protein studies are not very convincing and the conclusions in the discussion far too speculative.

The experimental work should be either restricted to the mRNA experiments, ideally extended to studies using human skeletal muscle cells. Then the study is an important contribution to theoretical risk of these vaccines. Or the protein studies need to be expanded substantially. Ideally mice should be vaccinated and the truncated proteins isolated from their plasma.

Abstract:

Lines 45-46: the authors should be very careful with the interpretation of VITT after mRNA vaccines. Detailed analyses of the clinical presentation of these patients indicates rather coincidence than causality.

Line 50: hnRNA should be explained

Introduction

Line 17: the numbers given are much too low, the UK has documented more than 300 cases, Germany has documented more than 150 cases. Either the authors should omit the numbers or provide a reference for the data are coming from.

Results

General format of all figures: figure legends are so small that they are already difficult to read in the provided figures, but will be impossible to read if the figures are printed in smaller size.

Page 5, lines 176-186: the data of luciferase activity in the medium are interesting. Can these data also be given for the codon optimized constructs used in the vaccines?

Figure 1C: I'm lost with this figure. There are many bands but how do the controls look like, for HEK cells, which are either not transfected to transfected with another construct? Apologies if I have not recognize the control. This is highly relevant as polyclonal antibodies as used for detection of the spike protein often show cross-reactivity with other proteins. Did the authors control the content of some of the bands by proteome analysis? According to the hypothesis of the authors the proteins found in the supernatant should be most relevant. According to the data showing in figure 1B very strong activity for the ChAdOx1 construct, one would have expected most truncated spike protein in the supernatant of ChAdOx1 transfected cells but this seems not to be the case, although the western blot is difficult to interpret.

Page 7: Experiments are very interesting as they show that splicing is highly dependent on the cell type. The vaccine is injected into the skeletal muscle. How do the findings obtained in HEK cells, HepG2 and murine C2C12 cells translate to what is happening in human skeletal muscle cells? Why were the experiments not performed in human skeletal muscle cell lines, when with the cell type and likely the species are so relevant? Do the transfected cell lines release truncated spike protein into the supernatant?

Discussion

Page 9 lines 371-373: the studies nicely show that splice reactions occur. However, the data that soluble Spike protein variants are generated are not very convincing. Especially as splicing is strongly cell type dependent, no evidence is provided for production of spliced, soluble spike protein variants by skeletal muscle cells.

The following part of the discussion is highly speculative. Activation of endothelial cells depends on the amount of secreted spike proteins. In addition it has been shown that after vaccination with mRNA vaccines spike protein is circulating in the vaccinated individuals. Endothelial cells carry the ACE receptor. Large amounts of the spliced spike protein but be needed to reach the cerebral veins and splanchnic veins without being completely diluted. When the blood is drained from the muscle where the majority of the vaccine is located it will first pass the lung before it can reach the splanchnic and cerebral vasculature. Why aren't all spike proteins captured by the receptors in the lung? If spike protein in the circulation is causing such severe cell activation the clinical studies with in activated virus or virus like particles expressing the spike protein should show massive adverse effects.

*Reviewer #2:*

The authors addressed a potentially third mechanisms by which vaccines against Sars-CoV2 (responsible for Covid-19), in particular DNA based vaccines could trigger rare forms of thrombosis. Their theory, for which they provide some initial evidence is that the vaccine induced transcript of the important and pathogenic viral "spike" protein would by alternative splicing ( a variant DNA transcription mechanism naturally occurring for many genes) produce spike protein that is not linked to a membrane (hence, localised) but becomes soluble and may circulate in blood to bind receptors that would otherwise recognise virus; the result may be triggering of immune mechanisms not by the virus, but by the spliced DNA products, the spike protein.

Although the data are reasonably straight forward, the story as such is hard to follow for the non informed reader (or clinician like me) and the data are preliminary as no evidence for pathogenicity of soluble spike proteins is presented.

The manuscript "Vaccine-Induced Covid-19 Mimicry Syndrome: Splice reactions in adenovirus-based current vaccines result in secretion of truncated Spike variants that may cause adverse side effects" by Kowarz and colleagues describes the potential splicing events and products of viral RNA originating from cloned vector DNA in the nucleus of eukaryotic cells. Overall, the description of results in the manuscript are reasonably clear for molecular biologists with an interest in virology, but very complex and difficult to follow for the more general reader of a journal like *eLife*. Their data provide evidence that splicing of RNA originating from vector DNA is possible and might lead to alternative spike proteins. There is no evidence presented demonstrating a direct relation between the alternative extracellular proteins and pathogenicity either in in vitro or an animal model. Thus, the statement on page 10 lines 380-381 "According to our results, soluble Spike variants may be an additional causative factor for the occurrence of rare but severe events observed after vaccination with Vaxzevria" isn't supported by the presented data. And why is this statement made for Vaxzevria only and not for the Janssen vaccine? Is there a different mechanism for the latter? Even the title is too speculative: the Spike variants may cause adverse side effects, but also may not….

Specific Comments:

Page 2, lines 57-59: "The Covid-19 pandemic, starting in the last months of 2019 in Wuhan (China) and caused by the RNA virus SARS-CoV-2, so far (19 Oct 2021) has resulted in more than 240 Mio infections and more than 4.93 Mio deaths." Please add a reference to this claim.

Page 2, lines 68-74: Please add references to these claims. In general, references are missing throughout the manuscript and various sentences can be interpreted as either fact or as speculation or proposing a hypothesis. For instance, on page 9 lines 374-475 "Obviously, severe life-threatening thromboembolic events can occur due to direct infection of recipient cells by wildtype SARS-CoV-2." Is this a fact or speculation?

Page 2, lines 72-74: "The most severe side effects entailed rare events of thrombocytopenia combined with cerebral venous sinus thrombosis (CVST) causing the patients' death in about one third of the cases." This statement might need some clarification as it is not clear if it is one third of the 150 cases, of the 6 cases or the overall 156 cases. Also, on third seems a lot but giving a percentage based on the >24 million doses given will change the picture dramatically.

Page 2, lines 77-78: "These thromboembolic events in combination with thrombocytopenia are related to another condition, "Heparin-induced thrombocytopenia" (HIT)." Are the thromboembolic events related to HIT or share similarities with HIT? A relation to HIT suggests that it might be causal related, which is absolutely not the case.

Page 3, lines 88-89: "In the rough endoplasmatic reticulum (ER) the mRNA is translated to become a membrane-anchored protein." Is the mRNA injected into the lumen of the rough ER and subsequently translated into protein or is the mRNA translated in the cytoplasm and the mRNA-ribosome-peptide docked to the ER membraned through the signal-recognition concept with translocation of the newly synthesized peptide across the ER membrane?

Page 4, lines 138-139: "Similarly, we performed the same in-silico analysis also for codon-optimized Spike open reading frames in three different adenovirus vector systems." The text following on this sentence could benefit from numbering the three DNAs.

Page 4, line 154: the term "unsaturated splice donor sites" might not be clear to the reader as it isn't explained in the previous part.

Page 5, lines 195-199: "medium supernatant of each investigated cell line displayed several protein bands migrating at molecular weights as predicted by the major splice events in each of the investigated cell lines (Ad5.S 195,1 and 195,3 kDa; ChAdOX1-S 195,1 and 195,3 kDa; Ad26.COV2.S 196,4 kDa; all without glycosylation)." The presence of protein bands in the medium supernatant needs clarification as before it is stated (line 179) that luciferase activity was detected intracellularly for all three adenoviral vector. What could be the origin of the extracellular signal in the medium supernatant?

In other words, it isn't clear what the meaning is of the part between lines 176 and 179: "We also cloned the three codon-optimized Spike genes of the adenoviral vectors (Ad5.S, ChAdOx1-S, Ad26.COV2.S) into the three versions of the luciferase splice trap (0/+1/+2) and analyzed the production of spike-luciferase fusion proteins. Here, we only analyzed intracellular luciferase activity (see Figure 1B, IV-VI)." If the codon optimized sequence is cloned before the luciferase splice trap, wouldn't this lead to RNA transcripts that can be spliced?

Pages 6 and 7: This is a complex part of the manuscript, at least to understand the meaning of the presented data. Initially, it is presented as a validation of the previous observed splicing of the DNA variants supplemented with various cell types. Isn't this a repetition and confirmation of the previous part? The introduction of the pIX and focus thereon, doesn't contribute to the readability of this part of the Results section.

---

## [Author Response]

Essential revisions:1) Please revise the discussion to address the comments about the protein studies (see reviewer 1 and 2) and add what further data would help to further substantiate the hypotheses presented in the manuscript.

We have addressed the reviewers’ comments and attenuated too strong claims as to the level of certainty, whether the observed protein products are involved in causing side effects following vaccination. In the discussion we have added a section, which further experiments would be helpful to substantiate the hypotheses presented in our manuscript.

2) Reviewer 2: "Their data provide evidence that splicing of RNA originating from vector DNA is possible and might lead to alternative spike proteins. There is no evidence presented demonstrating a direct relation between the alternative extracellular proteins and pathogenicity either in in vitro or an animal model. Thus, the claim on page 10 lines 380-381 "According to our results, soluble Spike variants may be an additional causative factor for the occurrence of rare but severe events observed after vaccination with Vaxzevria" isn't supported by the presented data. And why is this statement made for Vaxzevria only and not for the Janssen vaccine? Is there a different mechanism for the latter?" Please revise according to this comment.

We thank the reviewer for the comment. We have attenuated our statement to indicate that our data potentially could explain side effects, in particular observed with the AstraZeneca vaccine.

The posed problem is indeed a problem mainly of the AstraZeneca vaccine, as it has not been modified to reduce chances of cryptic splicing events. As summarized in Figure 4B, Janssen had modified the open reading frame of Spike to prevent distinct actions that occur within the Spike protein (furin cleavage, conformational change) and in addition, they have point-mutated all major splice sites. This did not completely prevent splicing, but splicing was strongly reduced.

3) Reviewer 1: "line 45-46: the authors should be very careful with the interpretation of VITT after mRNA vaccines. Detailed analyses of the clinical presentation of these patients indicates rather coincidence than causality." Please consider adding this.

We thank the reviewer for the comment. We have modified the sentence to make clear that VITT has been associated only with vaccination with adenovirus-based vaccine but not with mRNA-based vaccines.

4) Reviewer 1: "line 17: the numbers given are much too low, the UK has documented more than 300 cases, Germany has documented more than 150 cases. Either the authors should omit the numbers or provide a reference for the data are coming from." Please update the data and add a reference

These numbers in the manuscript were from the time when the manuscript was first written in April; we have now corrected and updated our manuscript accordingly, and have implemented Suppl Table S1 with current data from 23. December 2021.

5) Reviewer 2: Page 2, lines 72-74: "The most severe side effects entailed rare events of thrombocytopenia combined with cerebral venous sinus thrombosis (CVST) causing the patients' death in about one third of the cases." This statement might need some clarification as it is not clear if it is one third of the 150 cases, of the 6 cases or the overall 156 cases. Also, one third seems a lot but giving a percentage based on the >24 million doses given will change the picture dramatically." Please update the manuscript with data from recently published reference of the absolute risks, such as DOI: 10.1212/WNL.0000000000013148

These numbers in the manuscript were from the time when the manuscript was first written in April; we have now corrected and updated our manuscript accordingly, and have implemented Suppl. Table S1 with current data from 23. December 2021.

6) Please add references where missing for specific claims, as suggested by both reviewers.

We have added 2 missing references.

7) Reviewer 1: "General format of all figures: figure legends are so small that they are already difficult to read in the provided figures, but will be impossible to read if the figures are printed in smaller size." Please improve the figures to improve readability.

We have done so and formatted several figures accordingly.

Reviewer #1:This study investigates the transcription mechanism and resulting spike protein production of adenoviral vector DNA encoding the SARS CoV2 spike protein. The main finding is that the transcribed hnRNA is processed by the splice machinery of the human cell. This results in truncated constructs of the spike protein, which lose the transmembrane part and can be secreted into circulation. Splicing, however, is highly dependent on the cell type. Production of spliced spike proteins have been shown in the study using HEK cells, but the controls in this experiment a week. No evidence is given that human muscle cells can produce spliced spike protein. The authors show that a murine myoblast cell line can splice spike protein mRNA, but no data are given whether the murine cell line also produces and secretes the spliced protein. The authors speculate that these truncated forms of the spike protein bind to human endothelial cells and induce downstream biological effects. Taken together, the mRNA studies are interesting and hypothesis generating, the protein studies are not very convincing and the conclusions in the discussion far too speculative.

We thank the reviewer for the evaluation and the constructive critic.

In the text we have attenuated some of our conclusions to indicate the potentiality of our findings to explain (some of the) side effects.

As explained below, we have added in the discussion a paragraph on the future experiments, performed in animals, to formally address the question of pathogenetic relevance of our findings in vivo.

The experimental work should be either restricted to the mRNA experiments, ideally extended to studies using human skeletal muscle cells. Then the study is an important contribution to theoretical risk of these vaccines. Or the protein studies need to be expanded substantially. Ideally mice should be vaccinated and the truncated proteins isolated from their plasma.

We thank the reviewer for the suggestion. Although the main focus of our work has been at the RNA level, we also wished to demonstrate the sequelae of the splicing events at the protein level, since this bears the potential for side effects.

As the reviewers recommends, as the next step we plan to perform in vivo experiments in mice expressing the hACE2 receptor with the Spike variants as observed in our study to directly investigate potential side effects in an animal model. However, this work has been beyond the scope of this manuscript.

Abstract:Lines 45-46: the authors should be very careful with the interpretation of VITT after mRNA vaccines. Detailed analyses of the clinical presentation of these patients indicates rather coincidence than causality.Line 50: hnRNA should be explained

The term hnRNA has been replaced with mRNA, which is the appropriate term here.

IntroductionLine 17: the numbers given are much too low, the UK has documented more than 300 cases, Germany has documented more than 150 cases. Either the authors should omit the numbers or provide a reference for the data are coming from.

These numbers in the manuscript were from the time when the manuscript was first written in April; we have now corrected and updated our manuscript accordingly, and have implemented Suppl. Table S1 with current data from 23. December 2021

ResultsGeneral format of all figures: figure legends are so small that they are already difficult to read in the provided figures, but will be impossible to read if the figures are printed in smaller size.

Figures have been modified to make them more legible.

Page 5, lines 176-186: the data of luciferase activity in the medium are interesting. Can these data also be given for the codon optimized constructs used in the vaccines?

These luciferase reporter assay data have been done for the original Wuhan virus sequence, and also for the codon-optimized AZ and Janssen Spike sequences, as written in lines 176-186.

Figure 1C: I'm lost with this figure. There are many bands but how do the controls look like, for HEK cells, which are either not transfected to transfected with another construct? Apologies if I have not recognize the control. This is highly relevant as polyclonal antibodies as used for detection of the spike protein often show cross-reactivity with other proteins. Did the authors control the content of some of the bands by proteome analysis? According to the hypothesis of the authors the proteins found in the supernatant should be most relevant. According to the data showing in figure 1B very strong activity for the ChAdOx1 construct, one would have expected most truncated spike protein in the supernatant of ChAdOx1 transfected cells but this seems not to be the case, although the western blot is difficult to interpret.

This is exactly the point: the cell is producing so many Spike variants, which we detect with a specific polyclonal Spike serum, as well as with the specific Luciferase monoclonal antibody. What is shown in the paper is only the tip of an iceberg, as we also uploaded the total 12 western blots as well. At least from the bands with the highest molecular weight, we can estimate that they are derived from in-frame-fusions of major portions of Spike fused to the luciferase protein. The signals cannot be compared based on intensity, because the blots were exposed for different times in order to make the bands visible.

Page 7: Experiments are very interesting as they show that splicing is highly dependent on the cell type. The vaccine is injected into the skeletal muscle. How do the findings obtained in HEK cells, HepG2 and murine C2C12 cells translate to what is happening in human skeletal muscle cells? Why were the experiments not performed in human skeletal muscle cell lines, when with the cell type and likely the species are so relevant? Do the transfected cell lines release truncated spike protein into the supernatant?

We have used murine C2C12 cells because this cell line is the most commonly used standard muscle cell line that is currently used in research and development. We observed that independent of the cell line used, the very same cryptic splice sites were used, although efficiencies of splicing was variable. We would also like to mention, although not discussed in the manuscript, that it is well possible, according to own experiments performed in mice, that vector spreading following intramuscular injection can occur, so that also other cell types the skeletal muscle cells can be transduced.

DiscussionPage 9 lines 371-373: the studies nicely show that splice reactions occur. However, the data that soluble Spike protein variants are generated are not very convincing. Especially as splicing is strongly cell type dependent, no evidence is provided for production of spliced, soluble spike protein variants by skeletal muscle cells.The following part of the discussion is highly speculative. Activation of endothelial cells depends on the amount of secreted spike proteins. In addition it has been shown that after vaccination with mRNA vaccines spike protein is circulating in the vaccinated individuals. Endothelial cells carry the ACE receptor. Large amounts of the spliced spike protein but be needed to reach the cerebral veins and splanchnic veins without being completely diluted. When the blood is drained from the muscle where the majority of the vaccine is located it will first pass the lung before it can reach the splanchnic and cerebral vasculature. Why aren't all spike proteins captured by the receptors in the lung? If spike protein in the circulation is causing such severe cell activation the clinical studies with in activated virus or virus like particles expressing the spike protein should show massive adverse effects.

We only pose the possibility that secreted Spike variants may cause the thrombotic events. Future experiments will address exactly this issue (see above). Many papers have already shown that soluble Spike is highly dangerous and may cause thrombosis as well as thrombocytopenia, like the SARS-CoV-2 virus as well.

Reviewer #2:The authors addressed a potentially third mechanisms by which vaccines against Sars-CoV2 (responsible for Covid-19), in particular DNA based vaccines could trigger rare forms of thrombosis. Their theory, for which they provide some initial evidence is that the vaccine induced transcript of the important and pathogenic viral "spike" protein would by alternative splicing ( a variant DNA transcription mechanism naturally occurring for many genes) produce spike protein that is not linked to a membrane (hence, localised) but becomes soluble and may circulate in blood to bind receptors that would otherwise recognise virus; the result may be triggering of immune mechanisms not by the virus, but by the spliced DNA products, the spike protein.Although the data are reasonably straight forward, the story as such is hard to follow for the non informed reader (or clinician like me) and the data are preliminary as no evidence for pathogenicity of soluble spike proteins is presented.The manuscript "Vaccine-Induced Covid-19 Mimicry Syndrome: Splice reactions in adenovirus-based current vaccines result in secretion of truncated Spike variants that may cause adverse side effects" by Kowarz and colleagues describes the potential splicing events and products of viral RNA originating from cloned vector DNA in the nucleus of eukaryotic cells. Overall, the description of results in the manuscript are reasonably clear for molecular biologists with an interest in virology, but very complex and difficult to follow for the more general reader of a journal like eLife. Their data provide evidence that splicing of RNA originating from vector DNA is possible and might lead to alternative spike proteins. There is no evidence presented demonstrating a direct relation between the alternative extracellular proteins and pathogenicity either in in vitro or an animal model. Thus, the statement on page 10 lines 380-381 "According to our results, soluble Spike variants may be an additional causative factor for the occurrence of rare but severe events observed after vaccination with Vaxzevria" isn't supported by the presented data. And why is this statement made for Vaxzevria only and not for the Janssen vaccine? Is there a different mechanism for the latter? Even the title is too speculative: the Spike variants may cause adverse side effects, but also may not….Specific Comments:Page 2, lines 57-59: "The Covid-19 pandemic, starting in the last months of 2019 in Wuhan (China) and caused by the RNA virus SARS-CoV-2, so far (19 Oct 2021) has resulted in more than 240 Mio infections and more than 4.93 Mio deaths." Please add a reference to this claim.

Has been added with a date from December 2021.

Page 2, lines 68-74: Please add references to these claims. In general, references are missing throughout the manuscript and various sentences can be interpreted as either fact or as speculation or proposing a hypothesis. For instance, on page 9 lines 374-475 "Obviously, severe life-threatening thromboembolic events can occur due to direct infection of recipient cells by wildtype SARS-CoV-2." Is this a fact or speculation?

We have changed the introduction and all relevant references have now been cited.

Page 2, lines 72-74: "The most severe side effects entailed rare events of thrombocytopenia combined with cerebral venous sinus thrombosis (CVST) causing the patients' death in about one third of the cases." This statement might need some clarification as it is not clear if it is one third of the 150 cases, of the 6 cases or the overall 156 cases. Also, on third seems a lot but giving a percentage based on the >24 million doses given will change the picture dramatically.

We have changed the introduction completely and all relevant references have now been cited. We have also included the reference for this information, which is the publicly available safety surveillance data from the German Paul-Ehrlich Institute.

Page 2, lines 77-78: "These thromboembolic events in combination with thrombocytopenia are related to another condition, "Heparin-induced thrombocytopenia" (HIT)." Are the thromboembolic events related to HIT or share similarities with HIT? A relation to HIT suggests that it might be causal related, which is absolutely not the case.

We have modified the sentence to follow the reviewer’s suggestion.

Page 3, lines 88-89: "In the rough endoplasmatic reticulum (ER) the mRNA is translated to become a membrane-anchored protein." Is the mRNA injected into the lumen of the rough ER and subsequently translated into protein or is the mRNA translated in the cytoplasm and the mRNA-ribosome-peptide docked to the ER membraned through the signal-recognition concept with translocation of the newly synthesized peptide across the ER membrane?

We have changed the text accordingly to make this more understandable.

Page 4, lines 138-139: "Similarly, we performed the same in-silico analysis also for codon-optimized Spike open reading frames in three different adenovirus vector systems." The text following on this sentence could benefit from numbering the three DNAs.

We have rephrased the text to make it better understandable.

Page 4, line 154: the term "unsaturated splice donor sites" might not be clear to the reader as it isn't explained in the previous part.Page 5, lines 195-199: "medium supernatant of each investigated cell line displayed several protein bands migrating at molecular weights as predicted by the major splice events in each of the investigated cell lines (Ad5.S 195,1 and 195,3 kDa; ChAdOX1-S 195,1 and 195,3 kDa; Ad26.COV2.S 196,4 kDa; all without glycosylation)." The presence of protein bands in the medium supernatant needs clarification as before it is stated (line 179) that luciferase activity was detected intracellularly for all three adenoviral vector. What could be the origin of the extracellular signal in the medium supernatant?In other words, it isn't clear what the meaning is of the part between lines 176 and 179: "We also cloned the three codon-optimized Spike genes of the adenoviral vectors (Ad5.S, ChAdOx1-S, Ad26.COV2.S) into the three versions of the luciferase splice trap (0/+1/+2) and analyzed the production of spike-luciferase fusion proteins. Here, we only analyzed intracellular luciferase activity (see Figure 1B, IV-VI)." If the codon optimized sequence is cloned before the luciferase splice trap, wouldn't this lead to RNA transcripts that can be spliced?

Yes, exactly. The splice trap allows only to produce Luciferase activity when artificial splice events are occuring between the Spike open reading frame and the Luciferase open reading frame; most important: the luciferase gene cassette does not exhibit a start codon, so it is per se unable to produce Luciferase protein. If in-frame splicing occurs between the different reading frames of Spike proteins (Wuhan, Ad5.S, ChAdOX1.S, hAd26.COV2.S) and Luciferase, then Luciferase activity can be measured inside the cells (usually higher activity) and outside the cell (when secreted). Based on the knowledge of cell biology, secretion can only take place when the C-terminal membrane anchor becomes substituted by the Luciferase protein.

Pages 6 and 7: This is a complex part of the manuscript, at least to understand the meaning of the presented data. Initially, it is presented as a validation of the previous observed splicing of the DNA variants supplemented with various cell types. Isn't this a repetition and confirmation of the previous part? The introduction of the pIX and focus thereon, doesn't contribute to the readability of this part of the Results section.

We understand the reviewer’s concern. We have introduced small changes in the text. We believe, that this section is of high interest to researchers with interest in adenovirus vector development and gene therapy, since our observation bear significant relevance for any kind of adenovirus vector-based therapy, whether prophylactic as a vaccine or therapeutic in gene therapy. We resume the issue of read-through into the pIX gene and the consequences also in the discussion. Thus, the details provided in the Results section are also important for the discussion. Therefore, we would prefer to keep this important information in the manuscript.